# Layerwise Learning Rate in the Era of Large Language Models

## Abstract

Learning rate configuration is a fundamental aspect of modern deep learning. The prevailing practice of applying a uniform learning rate across all layers overlooks the structural heterogeneity of Transformers, potentially limiting their effectiveness as the backbone of Large Language Models (LLMs). In this paper, we introduce **Layerwise Learning Rate (LLR)**, an adaptive scheme that assigns distinct learning rates to individual Transformer layers. Our method is grounded in Heavy-Tailed Self-Regularization (HT-SR) theory, which characterizes the empirical spectral density (ESD) of weight correlation matrices to quantify heavy-tailedness. Layers with weaker heavy-tailedness are assigned larger learning rates to accelerate their training, while layers with stronger heavy-tailedness receive smaller learning rates. By tailoring learning rates in this manner, `LLR` promotes balanced training across layers, leading to faster convergence and improved generalization. Extensive experiments across architectures (from LLaMA to GPT-nano), optimizers (AdamW and Muon), and parameter scales (60M–1B) demonstrate that `LLR` achieves up to a 1.5× training speedup compared to uniform LR. Under the same training token budget, `LLR` further surpasses existing approaches by a clear margin. A key advantage of `LLR` is its low tuning overhead: it transfers nearly optimal LR settings directly from the uniform baseline, substantially lowering the barrier to practical adoption. Our code is submitted.

## 1 Introduction

Learning rate (LR) configuration is a cornerstone in modern deep learning, shaping both the convergence dynamics of training and the generalization ability of the resulting models (LeCun et al., 2015). Despite the rapid evolution of the LLM era—including the rise of Transformers (Vaswani et al., 2017), self-supervised learning (Radford et al., 2018), chain-of-thought reasoning (Wei et al., 2022), and RLHF (Ouyang et al., 2022)—the predominant LR strategy has remained essentially unchanged: applying a single LR value across all layers, which we refer to as the *Uniform LR*.

This paradigm, however, was originally developed for architectures with relatively homogeneous designs, such as multi-layer perceptrons (MLPs) and convolutional neural networks (CNNs). We contend that it does not adequately capture the heterogeneous and hierarchical nature of Transformer-based LLMs, thereby constraining their convergence and potentially their final performance. In particular, "Curse of Depth" (Sun et al., 2025) has revealed a distinct **hierarchical heterogeneity**, showing that the depthwise contributions of Transformer layers vary significantly. In addition, another line of work highlights the **architectural heterogeneity** of Transformers, where the Hessian spectra differ substantially across layer types (Zhang et al., 2024a; Wang et al., 2025; He et al., 2025). Together, these findings suggest that applying a single uniform LR might be inherently suboptimal, motivating the need for layerwise learning-rate strategies that explicitly account for both architectural and hierarchical heterogeneity in Transformer-based LLMs.

Research endeavors have explored the layerwise LR. For instance, the ratio between the weight norm and gradient norm of each layer has been used to set layerwise LRs, with the goal of stabilizing large-batch training in CNNs (You et al., 2017) and BERT (You et al., 2019). More recently, Wang et al. (2025) identified sharpness disparities across Transformer modules and introduced a fixed layerwise LR schedule determined via grid search. However, our preliminary evaluation in Figure 1 shows that these approaches are highly sensitive to LR tuning: their improvements emerge only

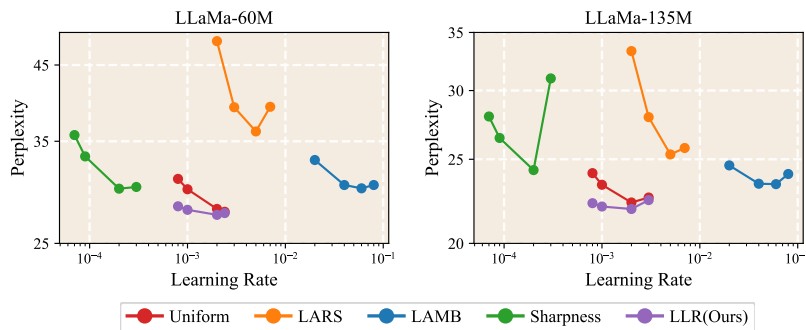

Figure 1: Learning rate sensitivity of different layerwise LR methods on LLaMa-60M and LLaMa-135M pre-training. Only the learning rate range with reasonable performance is reported for each method.

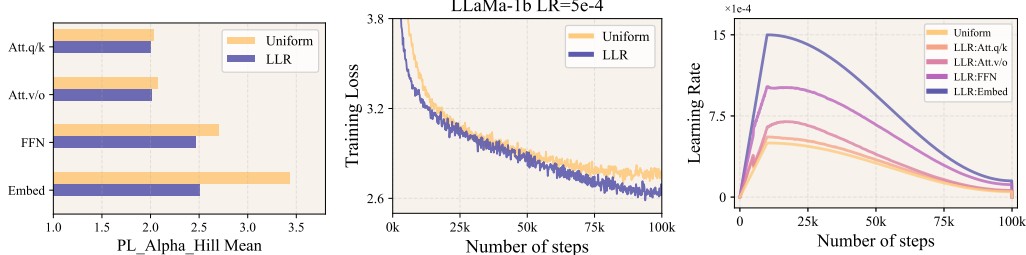

Figure 2: **Left:** Mean `PL_Alpha_Hill` value comparison (more balanced is preferred) between Uniform LR and LLR, with LLaMa-1B on C4; **Middle:** Training loss curves under the AdamW optimizer; **Right:** Layerwise learning rate schedules when combining `LLR` with cosine decay scheduler.

when compared against the uniform baseline at its suboptimal LR, while they often underperform the uniform baseline when the latter is tuned to its optimal LR. This leaves an open research question:

> *Does a principled layerwise LR scheme exist that can outperform the uniform LR even at its optimal LR?*

To this end, we propose **Layerwise Learning Rate (LLR)** for LLMs—an effective layerwise LR allocation strategy, grounded in Heavy-Tailed Self-Regularization (HT-SR) theory (Martin and Mahoney, 2019), that promotes balanced training across layers. `LLR` periodically measures the degree of heavy-tailedness in each layer's weight spectrum and assigns larger LRs to layers that are less heavy-tailed, while allocating smaller LRs to those exhibiting stronger heavy-tailedness. The underlying principle is drawn from HT-SR theory, which posits that layers with stronger heavy-tailedness are already better trained than those with weaker heavy-tailedness. Consequently, assigning larger LRs to the latter accelerates their training, thereby promoting more balanced progress across layers.

More concretely, `LLR` periodically computes the Empirical Spectral Density (ESD) across all layers, performs Power-Law (PL) fitting, and updates the corresponding `PL_Alpha_Hill` metrics. Layers with higher `PL_Alpha_Hill` values—typically embedding and FFN components—are assigned proportionally larger LRs, whereas layers with lower `PL_Alpha_Hill` values—such as attention modules—receive smaller LRs. This metric-to-map allocation scheme enables adaptive layerwise LR adjustment directly driven by spectral statistics, ensuring robust performance gains without extra hyperparameter-tuning burden. Our main contributions are as follows:

❶ We propose Layerwise Learning Rate (LLR), grounded by HT-SR theory, which enables dynamic assignment of LRs across layers within LLMs during training, promoting balanced training across layers.

❷ Extensive experiments on various architectures, including LLaMA to GPT-nano, optimizers with AdamW and Muon, parameter scales ranging from 60M to 1B, show that `LLR` yields up to a 1.5×

training speedup (Figure 6). Under the same training token budget, `LLR` further surpasses existing layerwise approaches by a clear margin (Table 2).

❸ A key advantage of `LLR` is its low tuning overhead: it inherits nearly optimal LR settings directly from the standard uniform LR (Figure 1), thereby lowering the practical barrier to adoption.

## 2 RELATED WORK

**Layerwise Learning Rate.** Although a uniform LR is the standard in deep network training, recent work has investigated layerwise LR allocation to improve optimization efficiency (Pan et al., 2024; Zhang et al., 2024b). LARS (You et al., 2017) and LAMB (You et al., 2019) scale LRs by the gradient-to-weight norm ratio, but as they were not designed for LLMs, they provide only marginal gains and demand additional tuning. Wang et al. (2025) introduced a blockwise LR allocation method that leverages sharpness disparities to accelerate pretraining, though it relies on exhaustive grid search. Complementarily, Hayou and Liu (2025) observed a dependence of optimal embedding-layer LRs on vocabulary size, suggesting distinct scaling rules across components. The closest related work is TempBalance (Zhou et al., 2023), which adapts HT-SR to adjust learning rates at the layer level for improved optimization. However, its applicability is restricted to CNNs and fine-tuning scenarios with limited data volume.

**HT-SR theory for LLM**. HT-SR theory describes a statistical phenomenon in which the weights of well-trained neural networks exhibit strong correlations, giving rise to heavy-tailed patterns in the ESD of Layerwise weight matrices (Mahoney and Martin, 2019; Martin et al., 2021). Empirical evidence shows that, across various stages of training—whether at the early, intermediate, or final phase—different components of LLMs (`Embed`, `Attention`, `FFN`) display distinct heavy-tailed structures in their ESDs (Couillet and Liao, 2022; Kothapalli et al., 2025; Ba et al., 2022). Building on this observation, numerous studies have sought to balance these heavy-tailed characteristics through applications of HT-SR, including model selection (Mahoney and Martin, 2019; Martin et al., 2021; Yang et al., 2023), module-wise adaptive training (Zhou et al., 2023), LLM pruning (Lu et al., 2024), and module-wise weight-decay allocation (He et al., 2025), achieving notable improvements in large-scale model training. However, it remains unclear if HT-SR can be utilized for layerwise LR designing for LLM pre-training.

## 3 METHODOLOGY

In this section, we revisit the HT-SR theory and outline the key metrics that underpin our analytical framework. Subsequently, we investigate the spectral characteristics of weights and gradients in LLM pretraining tasks, and, informed by these findings, we introduce the `LLR` algorithm, which utilizes the HT-SR theory to deliver substantial improvements in pretraining performance.

### 3.1 HT-SR THEORY

The HT-SR framework offers a systematic approach to characterizing the ESD of weight matrices in neural networks. Observations from prior work indicate that well-trained models tend to display ESDs with pronounced heavy-tails which is closely linked to higher training quality. Several studies have further revealed that, in LLMs, parameters that are well-optimized and those insufficiently trained can coexist throughout the entire training process, and such heterogeneity can significantly affect overall model performance (Zhou et al., 2023; He et al., 2025; Lu et al., 2024). Building upon this theoretical basis, we employ the HT-SR metric to measure the degree of spectral tail heaviness, applying lower LRs to layers with pronounced heavy-tail characteristics (e.g., `Att.q`, `Att.k`) and higher LRs to those with weaker heavy-tails (e.g., `FFN components`, `Embedding`), thereby promoting balanced optimization across layers to enhance generalization and overall effectiveness (see Figure 2). The extent of heavy-tailedness is determined quantitatively by fitting a power law (PL) to the ESD and using the resulting PL exponent ($\alpha$) as the measurement criterion.

We consider a network comprising $N$ layers, each associated with a weight or gradient matrix $\{W_l\}_{l=1}^{L}$ of shape $n \times m$ ($n \leq m$). For each layer, the ESD is computed from the eigenvalues of the correlation matrix $X_l = W_l^{\top} W_l$. Formally, the ESD is defined as $\text{ESD}(x) = \frac{1}{N} \sum_{i=1}^{N} \delta(x - \sigma_i)$, where $\delta(\cdot)$ denotes the Dirac delta function and $\sigma_i$ are the singular values.

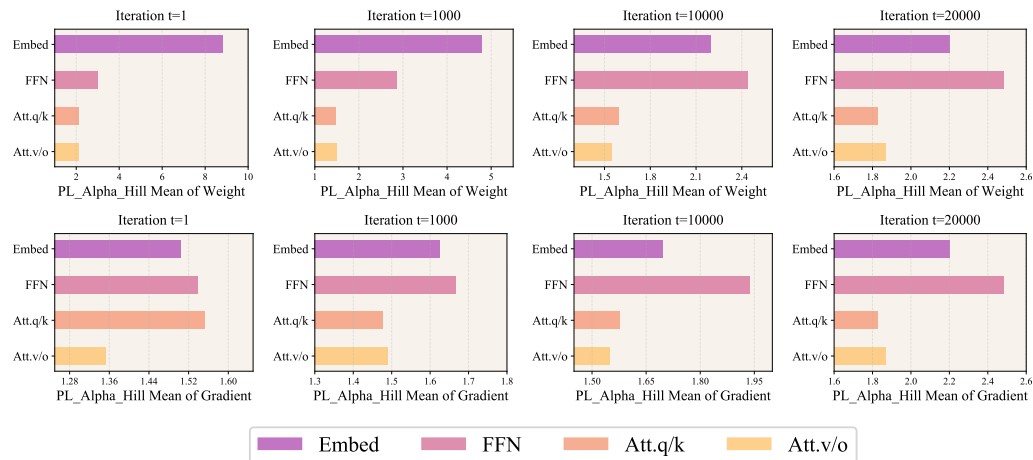

Figure 3: Evolution of `PL_Alpha_Hill` distributions from weights and gradients of LLaMA-135M components across iterations during pre-training on the C4 dataset, trained by AdamW with uniform LR.

We model the ESD using a power-law distribution of the form:

$$p(\lambda) \propto \lambda^{-\alpha}, \lambda_{\min} < \lambda < \lambda_{\max} \qquad (1)$$

where $p(\lambda)$ represents the eigenvalue density within the specified range, and $\alpha$ serves as a quantitative measure of the heavy-tailedness of the spectrum. The power-law exponent $\alpha$ is estimated using the Hill estimator (Zhou et al., 2023; Liu et al., 2024). Let $\{\lambda_i\}_{i=1}^{n}$ denote the eigenvalues sorted in ascending order. The Hill estimator is given by:

$$\texttt{PL\_Alpha\_Hill} = 1 + \frac{k}{\sum_{i=1}^{k} ln \frac{\lambda_{n-i+1}}{\lambda_{n-k}}} \qquad (2)$$

where the parameter $k$ controls the lower cutoff in the fitting process. In all experiments, we set $k = \frac{n}{2}$, thereby estimating the slope using the largest half of the eigenvalues (Detailed study of PL fitting method refer to Appendix C, Figure 8).

Figure 3 shows the `PL_Alpha_Hill` from the weights and gradients of LLaMA-135M components (`Embed`, `Attention`, `FFN`) at varying training iterations (t = 1, 1000, 10000, 20000). A pronounced and persistent layerwise disparity is observed throughout training: ❶ Across all iterations, `Embedding` and `FFN` layers consistently exhibit the highest `PL_Alpha_Hill` values, with `Embedding` far exceeding all other layers by a wide margin in the early stage; ❷ In early stages, weight-based `PL_Alpha_Hill` values remain largely stable, indicating that heavy-tailed characteristics in weight spectra have yet to develop and limiting their utility for guiding Layerwise LR allocation. In contrast, gradient-based `PL_Alpha_Hill` varies substantially from the outset, revealing pronounced component-level disparities consistent with prior works (Kothapalli et al., 2025; Ba et al., 2022); ❸ In later stages, `PL_Alpha_Hill` derived from weights and gradients converge across layers, reflecting similar degrees of heavy-tailedness and underscoring their joint utility as robust indicators of LLM training.

### 3.2 Layerwise Learning Rate for LLMs (LLR)

Based on the Layerwise `PL_Alpha_Hill` characteristics observed in Section 3.1, we propose a Layerwise LR (`LLR`) allocation method. `LLR` calculates the `PL_Alpha_Hill` values of all layers, assigning larger LRs to those with higher values and smaller LRs to those with lower values (see Figure 4). Notably, leveraging the temporal evolution of weight-based and gradient-based `PL_Alpha_Hill`, `LLR` uses the gradient-based metric for initialization to preserve sensitivity to Layerwise disparities. This gradient-driven allocation is applied during the first phase (up to $p=5\%$ tokens, detailed study refer to Appendix C, Figure 11) before transitioning to the weight-based metric to capture its matured heavy-tailed spectrum. The mapping from `PL_Alpha_Hill` values to per-layer LRs follows the bounded scaling function:

$$f_t(i) = \eta \cdot \left( \frac{\alpha_t^i - \alpha_t^{\min}}{\alpha_t^{\max} - \alpha_t^{\min}} (s-1) + 1 \right) \qquad (3)$$

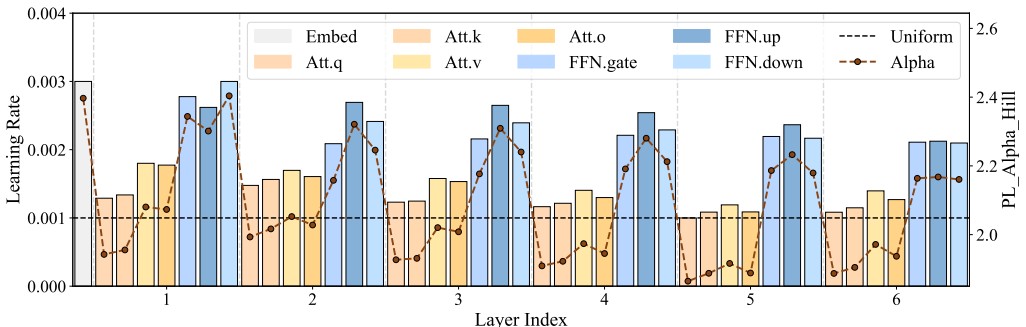

Figure 4: Given the imbalanced layerwise `PL_Alpha_Hill` of LLaMa-60M, `LLR` assigns lower LR to layers with lower `PL_Alpha_Hill`.

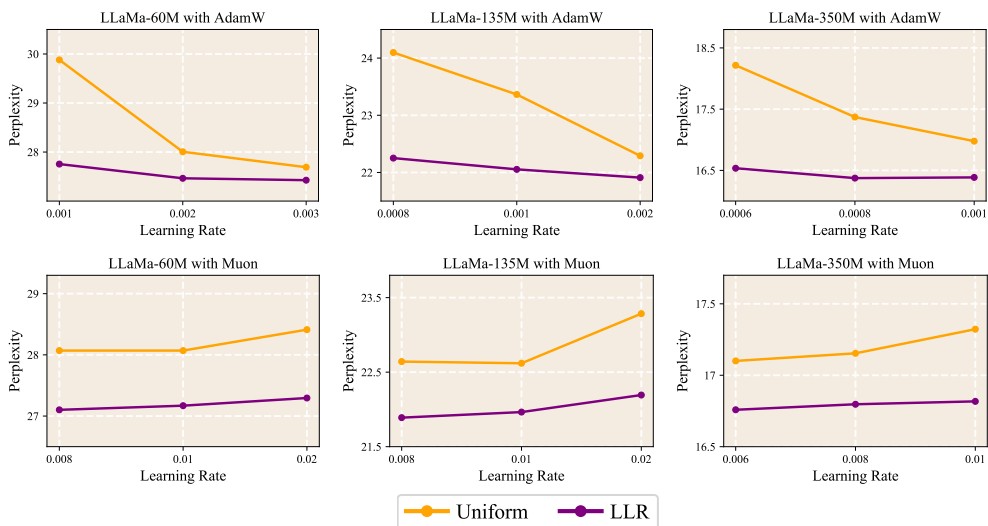

Figure 5: Validation perplexity for LLaMa models of varying sizes, comparing `Uniform` against `LLR`. The experiments employ two optimizers—AdamW (top row) and Muon (bottom row)—under three commonly adopted learning rates for each configuration.

where $\eta$ is the global LR, and $(1, s)$ define the range of scaling ratios applied to $\eta$. $\alpha_t^i$ is the `PL_Alpha_Hill` value of layer $i$ at step $t$, while $\alpha_t^{min}$ and $\alpha_t^{max}$ are the minimum and maximum `PL_Alpha_Hill` values among all layers at step $t$. Formula (3) guarantees that the adjusted LR, $f_t(i)$, remains within $[\eta, s\eta]$ as a scaled variant of $\eta$.

**Note that**, considering the `Embedding` layer's persistently high `PL_Alpha_Hill` values across all training stages, particularly in the initial phase, its LR is fixed at the scaling function's upper bound $s\eta$, as in (Wang et al., 2025), to avoid under-adaptation of the scaling function (3).

We provide the details of `LLR` in Algorithm 1. By integrating early-stage gradient-base sensitivity with later-stage weight-base stability, and by appropriately constraining dominant components, `LLR` establishes an adaptive and robust framework for layerwise LR allocation in LLM training. Extensive empirical studies across multiple model scales, optimizers, and LRs demonstrate that Layerwise LR allocation guided by `PL_Alpha_Hill` not only balances the `PL_Alpha_Hill` distribution (see Figure 2) but also consistently yields lower validation perplexity compared to the `Uniform` baseline (see Figure 5).

## 4 EMPIRICAL RESULTS

In this section, we begin by presenting the complete experimental setup (Section 4.1), followed by a comparison between `LLR` and several baselines (Section 4.2). Finally, we present our LLM pre-training results and the zero-shot/fine-tuning evaluation on commonsense reasoning tasks.

---

**Algorithm 1:** `LLR`

---

**Require:** Global LR $\eta$, number of training steps $T$, token proportion $p$ for gradient-driven `LLR`, interval $\tilde{t}$ of using `LLR`, maximum scaling ratio $s$, and $\alpha_t^i$ refers to $i_{th}$ layer's `PL_Alpha_Hill` at update step $t$

**for** $t \leftarrow 0$ *to* $T$ **do**

   **if** $mod(t, \tilde{t}) = 0$ **then**

      **Step 1:** Compute $\alpha_t^i$ for all layers using gradient-based Hill estimator (2) if $t/T \leq p$, otherwise weight-based Hill estimator;

      **Step 2:** Leverage $\alpha_t^i$ and adopt $f_t$ (3) to assign LR between $\eta$ and $s\eta$;

---

### 4.1 EXPERIMENTAL SETUP

**Models and Datasets**. We perform a comprehensive experimental study encompassing pre-training, zero-shot evaluation, and fine-tuning over a diverse range of model architectures and parameter scales. **(i) Pre-training.** Models are pre-trained on the C4 dataset (Raffel et al., 2020). We evaluate two categories of models: LLaMa-based architectures (60M, 135M, 350M, and 1B) and the GPT-nano architecture (130M). This design supports systematic evaluation of scalability and architectural generalization. **(ii) Zero-shot Commonsense Reasoning Evaluation.** Following pre-training, we evaluate the emergent reasoning capabilities of the LLaMa-1B checkpoints. This is conducted via zero-shot inference on a comprehensive benchmark suite of seven unseen commonsense reasoning tasks: PIQA (Bisk et al., 2020), SIQA (Sap et al., 2019), HellaSwag (Zellers et al., 2019), Wino-Grande Sakaguchi et al. (2021), ARC-c (Clark et al., 2018), ARC-e (Clark et al., 2018), and OBQA (Mihaylov et al., 2018). The evaluation utilizes the standard prompts provided by the lm-evaluation-harness to quantify out-of-data performance. **(iii) Fine-tuning.** We fine-tune the roberta-base model (Liu et al., 2019) on Commonsense Reasoning dataset (Hu et al., 2023) comprising the aforementioned seven downstream tasks, enabling a direct comparison of performance in a parameter-efficient adaptation scenario.

**Baselines.** We compare with four representative methods: (i) `Uniform`: Based on standard AdamW (Loshchilov and Hutter, 2017) or Muon (Liu et al., 2025) optimizers, applies the same learning rate to all layers without any layer-wise differentiation; (ii) `LARS` (You et al., 2017): A first-moment-based optimizer that assigns per-layer learning rates proportional to the ratio of weight to gradient norms; (iii) `LAMB` (You et al., 2019): A second-moment-based adaptive optimization algorithm that scales learning rates in a layer-wise manner according to weight norms to ensure training stability; (iv) `Sharpness` (Wang et al., 2025): Determines a fixed learning rate ratio across different layers via grid search based on sharpness information, without providing an automatic adjustment mechanism.

**Hyperparameters.** The detailed hyperparameter settings for all model sizes are summarized in Table 1 and Table 8-11 in Section B. All models are trained with gradient clipping at 1.0 and a cosine learning rate schedule, with 10% of the training tokens used for learning rate warmup. We conduct grid search over learning rates (see Table 1) and report the best configuration for each scale in the table. Learning rate settings and the corresponding $(1, s)$ parameter settings are also detailed in the tables. `LLR` is performed every 500 update steps throughout all experiments (Detailed study refer to Appendix C, Figure 9).

### 4.2 LLM PRE-TRAINING

In this section, we present the experimental results for `LLR` and all baselines across architectures (from LLaMa to GPT-nano), optimizers (AdamW and Muon), and parameter scales (60M–1B), followed by a comprehensive comparison between `Uniform` and `LLR`.

#### 4.2.1 MAIN RESULTS

Table 2 presents the main results of our study, where we evaluate the effectiveness of `LLR` and all baselines on the pre-training of LLaMa models with varying parameter scales (60M, 135M, 350M, and 1B) on the C4 dataset.

Table 1: Hyperparameters used in pre-training experiments. LLR-$s$ denotes the $(1, s)$ parameter setting in LLR.

| Model Size | Tokens | LARS-**LR/WD** | LAMB-**LR/WD** | Sharpness-**LR/WD** | AdamW/LLR-**LR/WD** | LLR-$s$ |
|---|---|---|---|---|---|---|
| 60M | 1B | 0.005/1e-6 | 0.05/0.1 | 0.0002/0.1 | 0.002/0.1 | 3 |
| 135M | 2B | 0.005/1e-6 | 0.05/0.1 | 0.0002/0.1 | 0.001/0.1 | 3 |
| 350M | 6B | 0.005/1e-6 | 0.05/0.1 | 0.0001/0.1 | 0.001/0.1 | 3 |
| 1B | 10B | 0.003/1e-6 | 0.05/0.1 | 0.0001/0.1 | 0.0005/0.1 | 3 |

Table 2: **(Main result).** Comparison with LLR and all baselines on pre-training various sizes of LLaMa models on C4 dataset. Validation perplexity (↓) is reported. All baselines are carefully tuned.

| Model Size | Uniform | LARS (You et al., 2017) | LAMB (You et al., 2019) | Sharpness (Wang et al., 2025) | LLR |
|---|---|---|---|---|---|
| 60M | 28.01 | 36.15 | 30.14 | 29.94 | **27.46** |
| 135M | 23.36 | 25.33 | 23.30 | 24.29 | **22.05** |
| 350M | 16.98 | 17.57 | 17.39 | 19.20 | **16.39** |
| 1B | 15.60 | 14.22 | 14.13 | 16.46 | **13.63** |

**Observations. ❶ Superior and consistent gains across all baselines.** Across all evaluated model sizes, LLR surpasses both the Uniform baseline and the adaptive learning rate methods (LARS, LAMB, Sharpness). This consistent superiority across various baselines strengths demonstrates the robustness of our approach and underscores its potential applicability in LLM pre-training. ❷ **Scalability to Larger Models.** The superiority of LLR is consistently observed from the smallest (60M) to the largest (1B) LLaMa models, achieving performance gains even at the billion-parameter scale. This consistency underscores the scalability and general applicability of our method in large-scale language model pre-training.

Furthermore, our experiments reveal that existing methods, originally designed for architectures without Attention components, such as LARS and LAMB, do not yield optimal results for LLMs. This may be attributed to their lack of consideration for the distinct characteristics and optimization requirements of Attention and FFN layers within transformer architectures. In contrast, our approach demonstrate that a tailored Layerwise LR allocation method can consistently enhance LLM training by explicitly accounting for the heterogeneous characteristics of different layers.

### 4.2.2 MORE MODEL STRUCTURES AND OPTIMIZER EXPERIMENTS

In this section, we further evaluate the proposed LLR across more model architecture and optimizer to demonstrate its generality and robustness.

**Results of GPT-nano.** We evaluate the proposed LLR on GPT-nano, as shown in Table 3. The comparison includes LARS, LAMB, Sharpness, Uniform (AdamW), and LLR. The results demonstrate that LLR achieves the lowest validation perplexity, significantly outperforming all baseline methods. This further confirms the robustness and broad applicability of our approach across different model architectures.

Table 3: **(GPT-nano Evaluation).** Performance comparison of LLR and all baselines on GPT-nano trained on the C4 dataset. Validation perplexity (↓) is reported, with all baselines carefully tuned.

| Method | Uniform | LARS | LAMB | Sharpness | LLR |
|---|---|---|---|---|---|
| Perplexity | 25.03 | 27.35 | 25.07 | 53.66 | **24.74** |

**Results with Muon Optimizer.** We investigate whether the proposed LLR can consistently improve performance when paired with optimizers beyond AdamW. Specifically, Table 4 presents results on various LLaMa model sizes trained with the Muon optimizer on the C4 dataset. Across all model scales, LLR delivers lower validation perplexity than Uniform, indicating that our method remains effective and complementary to alternative optimizers such as Muon.

Table 4: **(Pretraining with Muon).** Performance comparison on various sizes of LLaMa models trained with the Muon optimizer on the C4 dataset. Validation perplexity ($\downarrow$) is reported. The LR/WD format is: {LR for AdamW part} – {LR for Muon part} / {shared weight decay}. `LLR`-$s$ denotes the $(1, s)$ parameter setting in `LLR`. All configurations follow the same setup as described in Table 1.

| Model Size | Hyperparameters | | Validation Perplexity | |
|:---:|:---:|:---:|:---:|:---:|
| | LR/WD | `LLR`-$s$ | `Uniform` | `LLR` |
| 60M | 0.001-0.01/0.1 | 2 | 28.07 | **27.17** |
| 135M | 0.001-0.01/0.1 | 2 | 23.28 | **22.19** |
| 350M | 0.001-0.01/0.1 | 2 | 17.32 | **16.82** |
| 1B | 0.0006-0.006/0.1 | 2 | 14.50 | **13.79** |

Table 5: **(Up bound of LR scaling).** Validation perplexity ($\downarrow$) of AdamW or Muon with a uniform LR equal to $s$ times that in `Uniform` of Table 2 and Table 4, matching the maximum LR from `LLR`'s scaling method.

| Model Size | `Uniform`:AdamW | `LLR`:AdamW | `Uniform`:Muon | `LLR`:Muon |
|:---:|:---:|:---:|:---:|:---:|
| 60M | 130.84 | 27.46 | 28.41 | 27.17 |
| 135M | 22.59 | 22.05 | 23.61 | 22.19 |
| 350M | 16.67 | 16.39 | 24.49 | 16.82 |
| 1B | 14.50 | 13.63 | 14.15 | 13.79 |

**Performance of uniform LR at the scaling upper bound.** Table 5 presents the validation perplexity of different model sizes using AdamW and Muon optimizers with a uniform LR across all layers set to match the maximum Layerwise LR (upper bound) determined by `LLR`. Compared with the results in Table 2 and Table 5, `LLR` consistently outperforms both the upper bound and lower bound of uniform LR scaling across different optimizers and model sizes. These findings confirm that a uniform LR configuration is inherently suboptimal, whereas the proposed layer-wise LR strategy enables a more effective utilization of the capabilities of different optimizers.

### 4.3 MORE ANALYSIS

This section evaluates the downstream gains of all methods on zero-shot commonsense reasoning and finetuning tasks, and concludes with an analysis of the convergence speed improvements achieved by `LLR`.

**Zero-shot Results.** We evaluate the pretrained LLaMa-1B models, based on the checkpoints obtained from the pretraining experiments in Table 2, on 7 zero-shot commonsense reasoning tasks with `lm-eval-harness` under its default prompt configuration. As shown in Table 6, `LLR` achieves the best performance on 6/7 benchmarks. This indicates that the performance gains obtained from `LLR` during pretraining can transfer effectively to downstream reasoning tasks, highlighting its broad applicability.

Table 6: **(Zero-shot results of commonsense-reasoning tasks).** Zero-shot evaluation results ($\uparrow$) on seven commonsense reasoning benchmarks using the LLaMa-1B model pretrained with different methods.

| Method | OBQA | Winogrande | ARC-c | ARC-e | Hellaswag | SIQA | PIQA | Avg. |
|:---:|:---:|:---:|:---:|:---:|:---:|:---:|:---:|:---:|
| `Uniform` | 18.4 | 52.41 | 20.56 | 44.87 | 32.82 | 38.79 | 66.16 | 39.14 |
| `LARS` | 19.0 | 53.12 | 21.42 | 49.71 | 34.47 | 39.10 | 69.53 | 40.91 |
| `LAMB` | 20.0 | **54.14** | 20.82 | 50.63 | 34.48 | 39.71 | 68.50 | 41.18 |
| `Sharpness` | 17.8 | 51.93 | 19.37 | 45.92 | 31.16 | 37.26 | 65.56 | 38.43 |
| `LLR` | **20.8** | 52.72 | **21.90** | **51.43** | **35.80** | **39.87** | **70.08** | **41.80** |

**Finetuning Results.** We evaluate all methods on finetuning tasks using `roberta-base` from the Commonsense Reasoning dataset. As shown in Table 7, `LLR` achieves the best performance on

Table 7: **(Finetuning tasks).** Finetuning results on seven benchmarks from the Commonsense Reasoning dataset using `roberta-base` with different methods.

| Method | OBQA | Winogrande | ARC-c | ARC-e | Hellaswag | SIQA | PIQA | Avg. |
|---|---|---|---|---|---|---|---|---|
| Untuned | 16.0 | 48.61 | 22.52 | 26.34 | 23.52 | 35.05 | 50.54 | 31.80 |
| Uniform | 22.6 | 49.88 | 24.57 | 25.76 | 24.37 | 35.41 | 50.27 | 33.27 |
| LARS | 19.4 | 50.12 | 23.63 | 27.02 | 24.63 | 33.73 | 49.56 | 32.58 |
| LAMB | 16.0 | 48.62 | 23.12 | 26.68 | 23.14 | 36.08 | 50.05 | 31.96 |
| Sharpness | 21.8 | 49.01 | 23.21 | **25.13** | 24.25 | 35.47 | 51.03 | 32.84 |
| LLR | **23.6** | **51.30** | **24.59** | 25.21 | **24.92** | **36.24** | **51.20** | **33.86** |

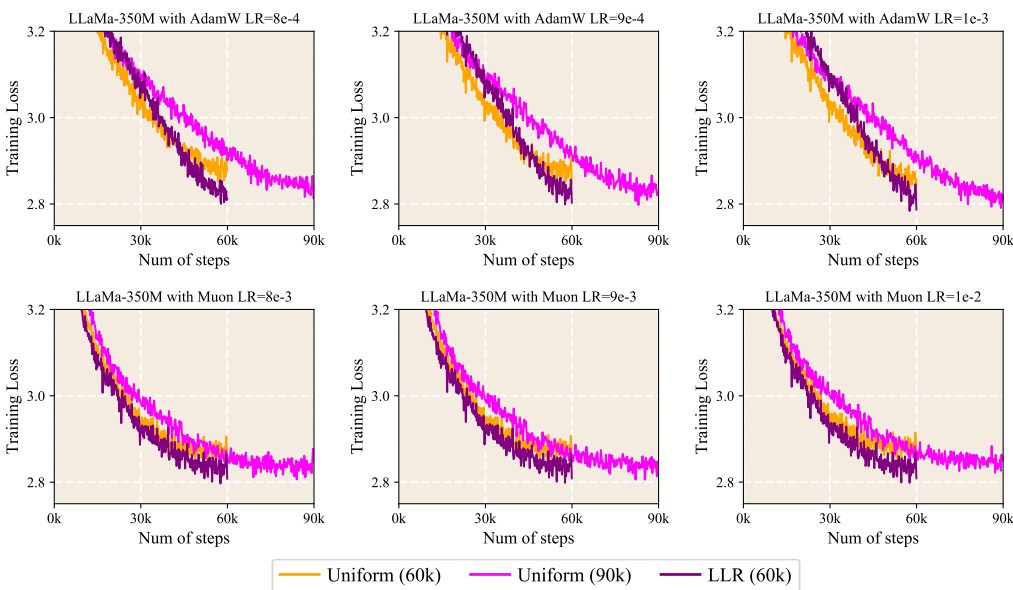

Figure 6: Training loss for the LLaMa-350M model, comparing `Uniform` (60k and 90k steps) against `LLR` (60k steps). The experiments employ two optimizers—AdamW (top row) and Muon (bottom row)—under three different learning rates for each configuration.

6/7 tasks. These results demonstrate that `LLR` is not only effective in the pretraining stage but also generalizes well to finetuning scenarios.

**Speedup.** Figure 6 presents the training loss curves for the LLaMa-350M model during pretraining, comparing `LLR` with `Uniform` across two optimizers (AdamW and Muon) and three learning rates. Across all configurations, `LLR` consistently achieves lower training loss with 60k steps compared to `Uniform` with the same number of training steps, highlighting its effectiveness in accelerating convergence. Furthermore, `LLR` attains performance comparable to or better than that of `Uniform` at 90k steps, corresponding to an approximate 1.5× speedup.

## 5 CONCLUSION

We introduced `LLR`, a layer-wise learning rate adjustment strategy that leverages `PL_Alpha_Hill` during LLM training. Across diverse setups, including different model architectures, multiple optimizers, and varying parameter scales, `LLR` consistently delivers lower perplexity and better downstream performance than existing baselines, while maintaining robustness to variations in training configurations. Moreover, `LLR` accelerates convergence by up to 1.5× across multiple optimizers. More importantly, it inherits near-optimal learning rate settings from standard uniform LR, minimizing tuning overhead and enabling easy integration into existing pipelines. These results suggest that `LLR` is an effective and broadly applicable optimization technique for LLM training.

**Limitations.** This work does not investigate the applicability of `LLR` to multimodal tasks, such as image-to-text or text-to-image generation, where optimization dynamics may differ substantially.

**Ethics Statement.** This research complies with the ICLR Code of Ethics. All model architectures used in our experiments, including LLaMa, GPT-nano and RoBERTa-base, are fully open source. The datasets, including C4 and Commonsense Reasoning benchmarks, are also publicly available and used in accordance with their licenses. No human or animal subjects were involved, and no personally identifiable or sensitive information was processed. We took care to minimize potential bias and ensure that our work raises no privacy or security concerns, while maintaining transparency and fairness throughout the study.

**Reproducibility Statement.** The paper and supplementary materials describe the training procedures, model setups, optimizer choices, and hardware details needed to replicate all experiments, and include the code used in our study. We hope these resources facilitate verification and encourage further exploration based on our findings.

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

# Appendix

## A    THE USE OF LARGE LANGUAGE MODELS

During the preparation of this manuscript, the use of LLMs was strictly confined to linguistic refinement tasks, including the correction of grammatical errors and the improvement of textual clarity and stylistic consistency. LLMs did not contribute to the conception of research ideas, the design or execution of experiments, the analysis or interpretation of results, or the generation of substantive scientific content.

## B    DETAILS OF EXPERIMENTS

This section provides detailed configurations for both pretraining and finetuning experiments. In section B.1, we present the architectural parameters of LLaMa and GPT-nano models, followed by the learning rate and weight decay settings for different methods. In section B.2, we describe the model and dataset parameters used for finetuning roberta-base on the Commonsense Reasoning Benchmark, along with the corresponding learning rate and weight decay settings.

### B.1    CONFIGURATIONS FOR PRETRAINING EXPERIMENTS

Table 8: Hyperparameters of LLaMa and GPT-nano models used in this paper.

| Models | Hidden | Intermediate | Heads | Layers | Steps | Data amount | Batch Size |
|--------|--------|--------------|-------|--------|-------|-------------|------------|
| LLaMa-60M | 768 | 2048 | 6 | 6 | 10K | 1B | 512 |
| LLaMa-135M | 768 | 2048 | 12 | 12 | 20K | 2B | 512 |
| LLaMa-350M | 1024 | 2736 | 16 | 24 | 60K | 6B | 512 |
| LLaMa-1B | 2048 | 5461 | 32 | 24 | 100K | 10B | 512 |
| GPT-nano | 768 | 3072 | 12 | 12 | 20K | 2B | 512 |

Table 9: Learning Rate and Weight Decay used for different methods in pretraining GPT-nano on C4 dataset.

| Method | Uniform | LARS | LAMB | Sharpness | LLR |
|--------|---------|------|------|-----------|-----|
| Learning Rate | 1e-3 | 5e-3 | 6e-2 | 3e-4 | 1e-3 |
| Weight Decay | 0.1 | 1e-6 | 0.1 | 0.1 | 0.1 |

### B.2    CONFIGURATIONS FOR FINETUNING EXPERIMENTS

Table 10: Hyperparameters used of all methods for finetuning roberta-base on the Commonsense Reasoning Benchmark.

| Model name | Hidden | Intermediate | Heads | Layers |
|------------|--------|--------------|-------|--------|
| Roberta-base | 12 | 3072 | 12 | 12 |

| Train Samples | Test Samples | Batch Size | Max_length | Training Epoch |
|---------------|--------------|------------|------------|----------------|
| 170K | 22.4K | 64 | 512 | 1 |

Table 11: Learning Rate and Weight Decay used for different methods in finetuning Roberta-based on the Commonsense Reasoning Benchmark.

| Method | Uniform | LARS | LAMB | Sharpness | LLR |
|--------|---------|------|------|-----------|-----|
| Learning Rate | 1e-4 | 5e-5 | 1e-4 | 1e-5 | 1e-4 |
| Weight Decay | 0.1 | 1e-6 | 0.1 | 0.1 | 0.1 |

## C   ABLATION EXPERIMENTS

This section presents ablation experiments evaluating the effects of different LR scheduling strategies, metrics, and related factors. We begin with repeated experiments across multiple random seeds (Table 12), with statistical validation via dependent t-tests. Table 13 demonstrates the robustness of `LLR` when evaluated across varying sequence lengths. Figure 7 compares scheduling metrics, analyzing validation perplexity and runtime. Figure 8 examines the impact of different PL fitting methods on model performance. Figure 9 investigates the impact of varying PL fitting gaps on model performance. Figures 10 and Figure 11 investigate the effects of varying the $(1, s)$ hyperparameter and the weight-to-gradient ratio with different PL fitting gaps in `LLR`.

**Repeat Experiments.**   Table 12 provides a comparison of several LR scheduling strategies, evaluated through repeated experiments with different random seeds. The dependent t-test results further substantiate these findings, with statistically significant p-values supporting the superiority of `LLR` over all other optimizers.

Table 12: **(Dependent t-test results on C4 with LLaMA-135M using the AdamW optimizer).** Each method is evaluated over six repeated experiments with random seeds $\{5, 6, 7, 8, 9, 10\}$, and compared against `LLR` using a dependent t-test. Perplexity is reported as mean $\pm$ standard deviation. The resulting p-values correspond to comparisons with `LLR`.

| Method | Uniform | LARS | LAMB | Sharpness | LLR |
|---|---|---|---|---|---|
| Perplexity | $23.33 \pm 0.05$ | $25.53 \pm 0.17$ | $23.36 \pm 0.04$ | $24.51 \pm 0.27$ | $22.11 \pm 0.04$ |
| P-value | 2.1e-12 | 1.2e-8 | 2.3e-13 | 2.3e-6 | |

**Varying Sequence Length.**   We examine the performance of all methods under different sequence lengths during pretraining. As shown in Table 13, using the `LLaMa-135M` architecture on the C4 dataset, `LLR` consistently achieves the best results across all sequence length settings $\{256, 512, 768, 1024\}$, demonstrating its robustness to variations in sequence length.

Table 13: **(Varying Sequence Length).** Pretraining results using `LLaMa-135M` with different methods under varying sequence lengths $\{256, 512, 768, 1024\}$. Validation perplexity ($\downarrow$) is reported.

| Seq-Length | Uniform | LARS | LAMB | Sharpness | LLR |
|---|---|---|---|---|---|
| 256 | 23.36 | 25.33 | 23.30 | 24.29 | **22.05** |
| 512 | 22.19 | 24.64 | 21.96 | 22.68 | **20.91** |
| 768 | 21.75 | 24.19 | 21.48 | 22.11 | **20.49** |
| 1024 | 22.03 | 24.39 | 21.55 | 22.30 | **20.53** |

**Varying HT-SR metrics.**   To investigate the effect of different learning rate scheduling metrics on model performance, we conducted ablation studies comparing these methods under three fixed learning rates. While prior work has primarily explored `Uniform` scheduling and `GradNorm` as representative metrics, our study additionally evaluates `PL_Alpha_Hill`, `FrobeniusNorm`, and `SpectralNorm` under identical training settings. Results in Figure 7 show that, across most learning rates, `PL_Alpha_Hill` consistently achieves the lowest validation perplexity (lower is better), highlighting its effectiveness over other evaluated metrics.

**Varying PL fitting methods.**   In our proposed framework, the HT-SR metric `PL_Alpha_Hill` is derived through PL fitting, and the choice of fitting method can affect both computational efficiency and final training effectiveness. To assess this impact, we evaluate `Goodness-of-fit` (Alstott et al., 2014; Martin et al., 2021; Clauset et al., 2009), `Fix-finger` (Yang et al., 2023), and `Median` (Zhou et al., 2023) under identical experimental conditions in Figure 8. Across all tested learning rates, `Median` not only maintains competitive or superior training performance but also substantially reduces computation time compared to the other two approaches, making it the preferred choice for PL fitting within our method.

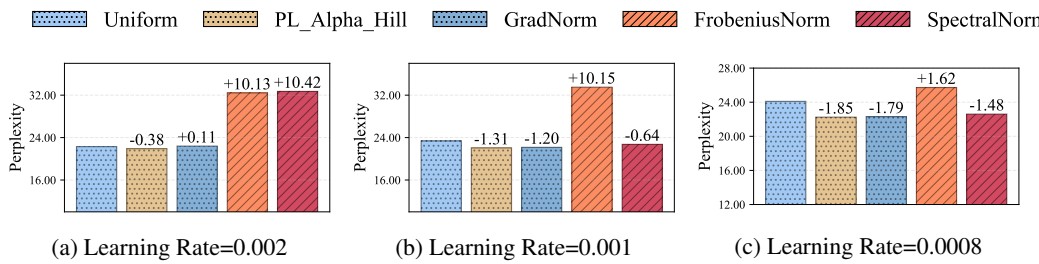

Figure 7: **(Varying HT-SR metrics).** Comparing `PL_Alpha_Hill` with multiple metrics under different learning rate settings. All experiments are conducted on LLaMa-135M.The value on the top of each bar indicates the difference from the leftmost bar in each plot and the same processing is applied in Figure 8, Figure 9, Figure 10 and Figure 11.

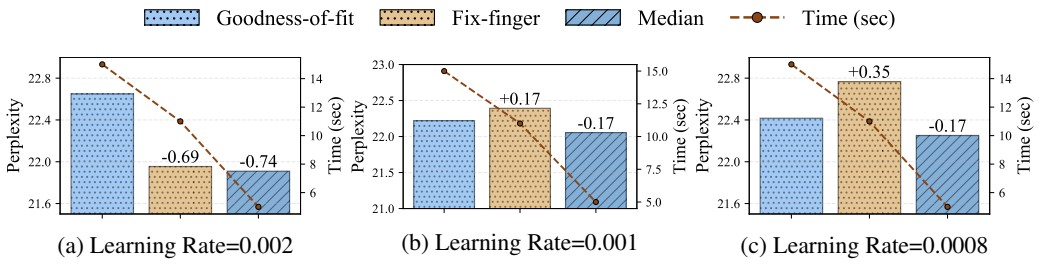

Figure 8: **(Varying PL fitting methods with different learning rates).** Analysis of three PL fitting methods—`Goodness-of-fit`, `Fix-finger`, and `Median`—across multiple learning rate settings. The bar plot and left y-axis represent perplexity (lower the better), while the line plot and right y-axis indicate the time taken for `LLR` once (in seconds, lower the better). The computation times are averaged over all PL fitting operations throughout the model training process. All experiments are conducted using LLaMa-135M.

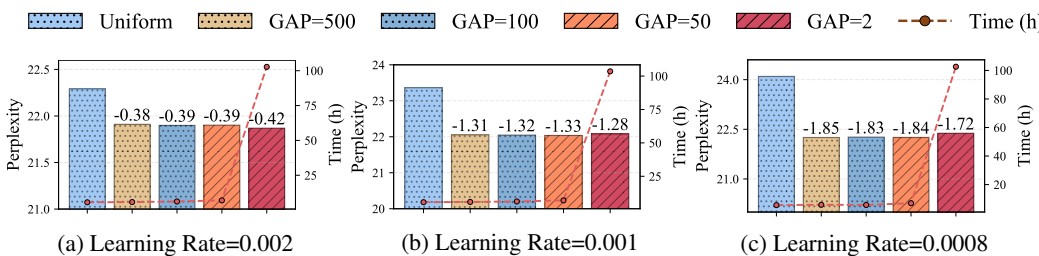

Figure 9: **(Varying PL fitting gaps).** PL fitting is conducted at varying gaps across training steps to evaluate the trade-off between performance and efficiency. Bars represent validation perplexity (lower is better), and overlaid lines show the total training time, measured in NVIDIA A100 GPU hours. All experiments are performed using the LLaMa-135M model.

**Varying PL fitting gaps.** To evaluate the effect of PL fitting frequency on model performance and computational efficiency, we compare different update gaps under identical experimental conditions in Figure 9. Across all tested learning rates, our method achieves stable performance across all gap settings, consistently outperforming the uniform baseline. Even when the fitting interval is as large as 500 training steps, the method maintains competitive perplexity while significantly reducing training time, demonstrating robustness and computational efficiency. These results justify using a larger fitting gap in practice to balance performance and cost.

**Hyperparameter study on** $(1, s)$**.** Figure 10 demonstrates that `LLR`, with $(1, s)$ settings of $(1, 2)$, $(1, 3)$, and $(1, 4)$, consistently surpasses the `Uniform` baseline across all learning rate configura-

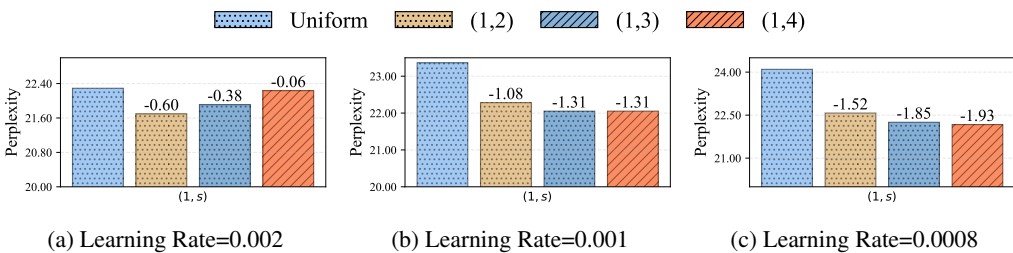

Figure 10: **(Hyperparameter study on $(1, s)$).** Results of a hyperparameter search for $(1, s)$ across different learning rate settings on the C4 dataset. The bar plots display validation perplexity (lower is better). All experiments are conducted using the LLaMa-135M architecture.

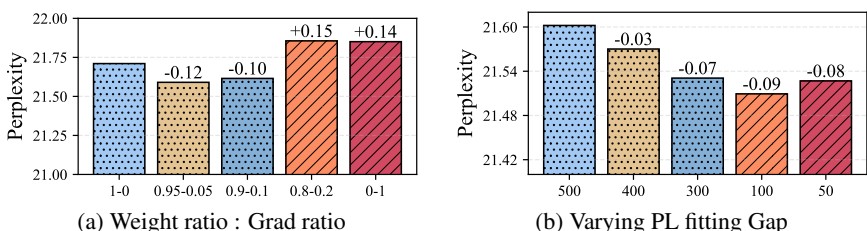

Figure 11: **Effects of weight-to-gradient ratio and PL fitting gap in `LLR`.** Pretraining results of the `LLaMa-135M` model on the C4 dataset when varying (a) the ratio of weight to gradient used in computing `PL_Alpha_Hill`, and (b) the update frequency of the layer-wise learning rate when using gradients to compute `PL_Alpha_Hill` (the update frequency for weights is fixed at every 500 update steps). Validation perplexity ($\downarrow$) is reported.

tions, with stable gains across schedules, highlighting its robustness and insensitivity to reasonable variations in $s$.

**Effects of weight-to-gradient ratio and PL fitting gap in `LLR`.** We further evaluate the effect of different hyperparameter choices within `LLR`. As shown in Figure 11, in subfigure (a), using gradients instead of weights to compute `PL_Alpha_Hill` during the initial 5%-10% of the total training steps leads to consistent performance improvements. In subfigure (b), when gradients are used to compute `PL_Alpha_Hill`, increasing the frequency of layer-wise learning rate updates during this initial phase also yields stable and consistent gains.

# D    REBUTTAL SUPPLEMENT

## D.1    LAYER-WISE LEARNING RATE DYNAMICS

To better understand how `LLR` adapts the optimization process over the course of training, we analyze the evolution of layer-wise learning rates in LLaMA-60M. In particular, we track the learning rates of all major parameter groups at different optimization steps.

Figure 12 reports the layer-wise learning rate distribution of all parameter groups at several representative training stages when training LLaMA-60M with `LLR`. In the early phase of training (roughly the first 10–20% of steps), the layer-wise learning rates exhibit pronounced variation across layers and parameter types. This indicates that `LLR` aggressively reshapes the effective optimization landscape at the beginning of training, assigning relatively larger or smaller learning rates to different layers as needed to accelerate convergence and stabilize the model.

After this initial phase, the layer-wise learning rates become much more stable, and their changes over the remaining training steps are relatively small. The distribution of learning rates across layers gradually converges to a steady pattern, suggesting that `LLR` transitions from an exploratory adjustment regime to a more conservative regime where only minor refinements are made. This be-

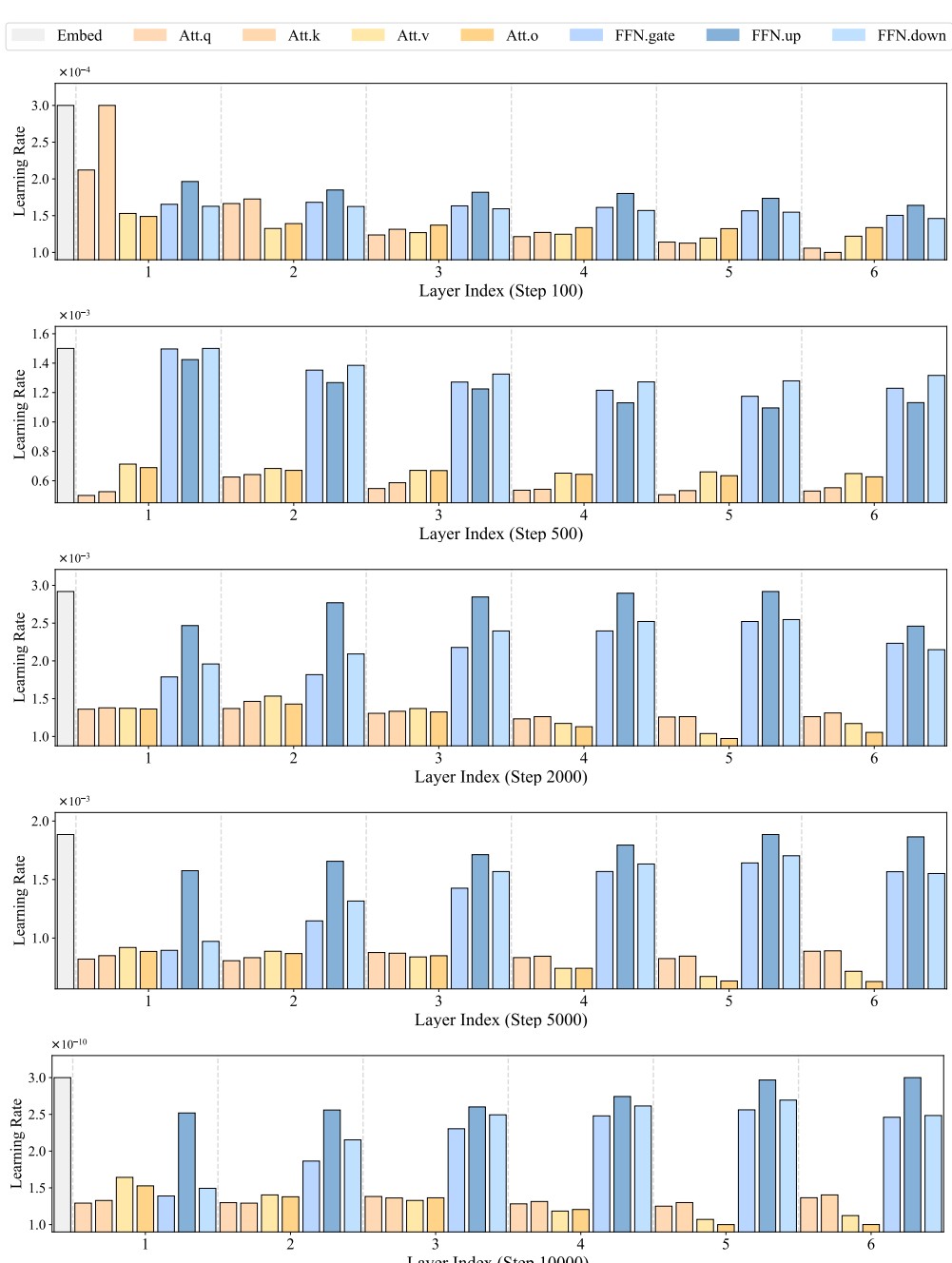

Figure 12: Layer-wise learning rate schedules of all parameter groups when training LLaMA-60M with `LLR` at different optimization steps.

havior implies that most of the critical layer-wise adaptation happens early in training, while the later stages primarily focus on fine-tuning the learned representations under a nearly stationary learning rate configuration.

## D.2 LAYER-WISE PL_ALPHA_HILL DYNAMICS

In this section, we investigate how `LLR` influences the optimization dynamics of different parameter groups, we analyze the evolution of the `PL_Alpha_Hill` in LLaMA-130M. Figure 13 shows the `PL_Alpha_Hill` for each major parameter group when training with `LLR` (perplexity = 22.05) and `Uniform` (perplexity = 23.36). We compute `PL_Alpha_Hill` every 500 up-

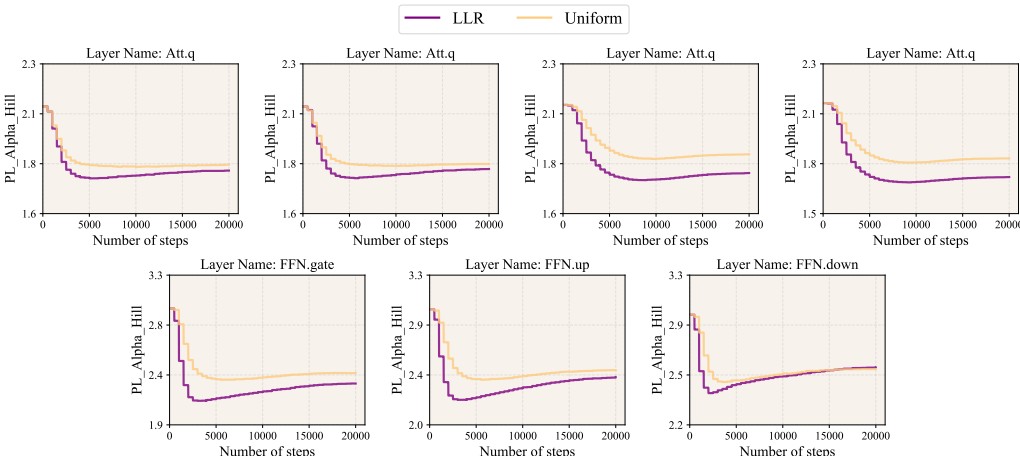

Figure 13: Evolution of `PL_Alpha_Hill` for different parameter groups when training LLaMA-130M with `LLR` (perplexity = 22.05) and `Uniform` (perplexity = 23.36). The curves are computed every 500 training steps.

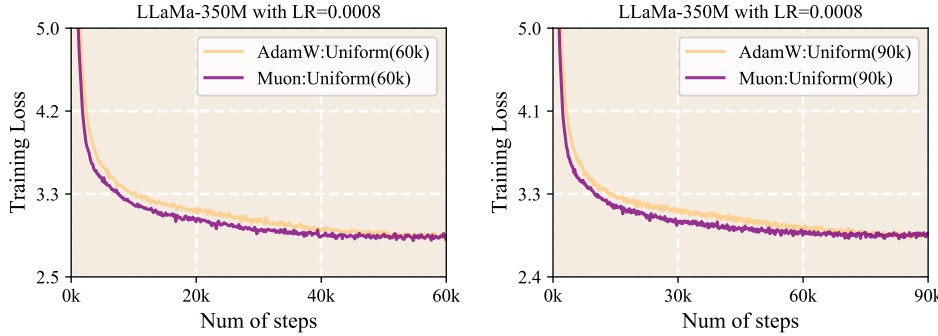

Figure 14: Training loss curves of LLaMA-350M with LR = 0.0008 under different optimizers. We compare AdamW and Muon using uniform learning rates for 60k (left) and 90k (right) training steps.

date steps. Several trends are evident from the figure.❶ The `Attention` parameters (`Att.q`, `Att.k`, `Att.v`, `Att.o`) consistently exhibit smaller `PL_Alpha_Hill`, whereas the `FFN` parameters (`FFN.gate`, `FFN.up`, `FFN.down`) maintain larger `PL_Alpha_Hill` throughout training. ❷ The `PL_Alpha_Hill` of all parameter groups change substantially during the early phase of training (approximately the first 20% of update steps). This pronounced variation highlights the importance of periodically updating the layer-wise LRs, rather than keeping them fixed over time. ❸ Compared with `Uniform`, `LLR` markedly reduces the `PL_Alpha_Hill` across all parameter groups. This reduction correlates with the improved perplexity achieved by `LLR`, suggesting that better control of `PL_Alpha_Hill` contributes directly to the enhanced training performance.

### D.3 OPTIMIZATION DYNAMICS OF ADAMW AND MUON

In this subsection, we examine how different optimizers affect the pre-training dynamics with LLaMA-350M. As shown in Figure 14, at both training budgets we observe distinct optimization behaviors: (1) with AdamW, the rapid loss decrease mainly occurs in the later stage of training; (2) in contrast, Muon achieves a faster loss reduction in the early stage of training, leading to consistently lower training loss than AdamW throughout most of the trajectory.

As shown in Figure 15, for AdamW the Uniform(60k) and Uniform(90k) curves do not intersect at 60k update steps because AdamW exhibits a rapid loss drop in the later stage of training: by 60k steps, the 60k-run has already completed this late-stage descent, while the 90k-run has not yet entered it, so the loss of Uniform(60k) remains lower than that of Uniform(90k). In contrast, for Muon the Uniform(60k) and Uniform(90k) curves intersect around 60k steps because Muon's rapid

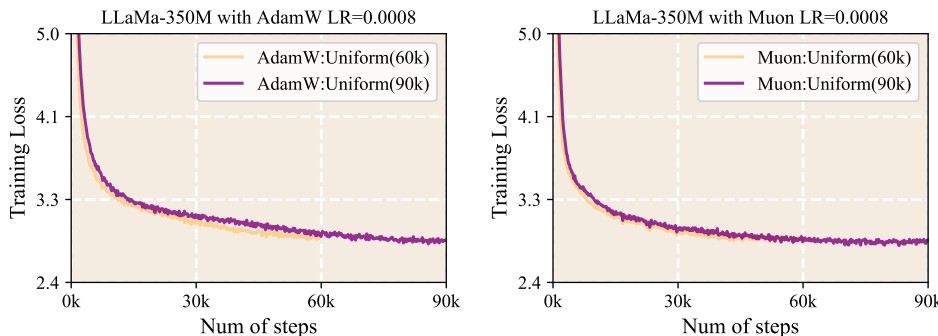

Figure 15: Training loss of LLaMA-350M with LR = 0.0008 under AdamW (left) and Muon (right), comparing uniform training budgets of 60k and 90k steps.

loss reduction occurs in the early stage of training; by 60k steps, the 90k-run has already finished this early sharp descent and maintains a lower loss, causing the two curves to cross.

## D.4 ROBUSTNESS OF LLR TO HEAD DIMENSION

In this subsection, we investigate how the attention head dimension affects the behavior of `LLR` during pre-training. We train LLaMA-350M with AdamW under several head dimensions and measure the resulting `PL_Alpha_Hill` values across different attention and feed-forward components. As shown in Figure 16, varying the head dimension has only a minor impact on the `PL_Alpha_Hill` values: the curves remain nearly flat across head sizes, highlighting the stability of the `LLR` method with respect to this architectural change.

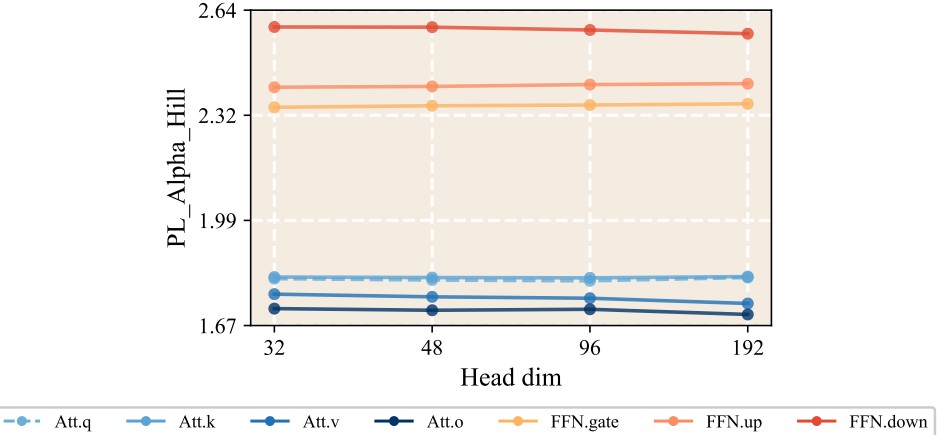

Figure 16: `PL_Alpha_Hill` values of LLaMA-350M trained with AdamW under different head dimensions.