# OpenReview forum: "Layerwise Learning Rate in the Era of Large Language Models"
_ICLR.cc/2026/Conference — Submitted to ICLR 2026_

### Official Review · Reviewer_uFmm · 2025-10-20

**Soundness:** 2
**Presentation:** 4
**Contribution:** 2
**Rating:** 4
**Confidence:** 4

**Summary:**

This paper proposes Layerwise Learning Rate (LLR), an adaptive strategy that assigns distinct learning rates to Transformer layers based on Heavy-Tailed Self-Regularization (HT-SR) theory. The method measures each layer’s spectral heavy-tailedness via the `PL_Alpha_Hill` metric and scales learning rates. The authors demonstrate that this approach balances optimization across layers, accelerates convergence, and improves generalization, with comprehensive experiments.

**Strengths:**

- The paper is well-structured and easy to follow, demonstrating clear organization and logical flow.
- Conceptually interesting: operationalizing HT-SR signals as a dynamic LR allocator; the gradient-→weight metric handoff is a reasonable design to handle HT-SR.
- Broad empirical sweep: multiple model sizes (60M–1B), two optimizers (AdamW, Muon), distinct architectures (LLaMA, GPT-nano), plus downstream zero-shot and fine-tune results; helpful ablations on metric choice, fitting frequency, and $(1,s)$ sensitivity.

**Weaknesses:**

My primary concerns is presented in questions, my other concerns:
- **Limited Theoretical Depth.** While HT-SR provides motivation, the derivation of the mapping from spectral exponent to learning-rate scaling remains heuristic and empirical rather than theoretically grounded.
- **Comparative Baseline Weakness.** Competing methods like LARS, LAMB are rarely used in LLM context, comparative baselines are generally weak.
- **Method Naming** Layerwise Learning Rate is a widely used generic term in DL; it is not suitable as a distinctive method name. A more distinctive name would avoid confusion.

**Questions:**

**Crucial Questions**

- **Scalability to Larger Scales.** Most of experiments are below 350M model size. There is one experiemnt with 1B model size. Moreever, the training token in this experiment is only 10B, even below the Chinchilla law, which makes reader doubting aboout its scalability. Can the authors provide larger-scale runs or scaling-law extrapolations to demonstrate robustness beyond 1 B parameters?
- **Sensitvity of hyperparameter $s$** Let loss $f$ be a function of $s$, as long as $0$ is not the minimizer of $f$, one can find a better $s$ with lower loss. The work chooses different $s$ such as 2.0 and 3.0. Please clarify the selection criterion for $s$, its sensitivity range, and whether optimal $s$ scales with model depth or spectral spread
- **Relation to Other Works** The relation with work [1] and [2] needs to be clarified.
  - I examined [1] (AlphaDecay) detailedly, AlphaDecay also leverages HT-SR theory to develop a layerwise weight decay method. How could HT-SR inspire two highly similar methods—one for weight decay, one for learning rate? First, The authors need to further clarify the relations with work [1]. Second, Which way is better, LLR or AlphaDecay? Is LLR orthogonal to AlphaDecay? Do gains add? (BTW, The paper's images are very similar with these of [1].)
  - [2] performs grid search to get the best scaling $[1, 8, 4, 6, 10]$ on their settings to show the consistency with their principle. First, as you tune $s$, can you run experiment with the $s$-parametrized scaling $[1, 1+7s/9, 1+3s/9, 1+5s/9, 1+s]$ as comparison, other setting remains the same as LLR. Second, can you perform a grid search, to show that the best scaling in your setting is more close to your HT-SR or their principle?
- **Suspicious Results** Can you explain these results?
  - Figure 6 presents six images, but does not provide enough information. Moreever, Under Muon, the losses of Uniform (60k) and Uniform (90k) look similar near step 60k, yet Uniform (60k) has already finished cosine decay by then in some subplots; if so, it should be lower.
  - On Table 2, the ppl gap between `Uniform` and `LLR` for model size 60M, 135M, 350M, 1B is 0.55, 1.31, 0.59, 1.97. As model size grows, the baseline perplexity decreases, so further reduction should be increasingly difficult [3]. The large gap at 1B seems abnormal and warrants verification of hyperparameter parity and token-budget consistency.

Other questions are presented in the other concerns of Weaknesses. How these concerns are addressed will greatly affect my score.


[1] He, D., Jaiswal, A., Tu, S., Shen, L., Yuan, G., Liu, S. and Yin, L., 2025. AlphaDecay: Module-wise Weight Decay for Heavy-Tailed Balancing in LLMs. NeurIPS 2025. \
[2] Wang, J., Wang, M., Zhou, Z., Yan, J. and Wu, L., 2025. The sharpness disparity principle in transformers for accelerating language model pre-training. ICML 2025. \
[3] Wen, K., Hall, D., Ma, T. and Liang, P., 2025. Fantastic pretraining optimizers and where to find them. arXiv preprint arXiv:2509.02046.

---

> ### Author Response · Authors · 2025-11-22
>
> >**Q1.Limited Theoretical Depth. While HT-SR provides motivation, the derivation of the mapping from spectral exponent to learning-rate scaling remains heuristic and empirical rather than theoretically grounded.**
>
> **A1:** Thank you for this thoughtful comment about the theoretical depth of our method. Our focus is on extending and validating these ideas empirically in LLMs, where we show that α can be used as a useful signal for layerwise LR scaling.
>
> The formal connection between the power law (PL) exponent α of weight matrices and model capacity/generalisation is already established in HT-SR literature [1–3]. These works show that α provides a principled, theoretically motivated measure of effective capacity and implicit regularisation. In our paper, we therefore deliberately avoid “reinventing the wheel” by re deriving these results, and instead build on this established theory. In the revision, we will add a concise “theory recap” in the method section and explicitly cite the key theorem from HT SR to make this link clearer.
>
> [1] Martin, C. H., Peng, T., & Mahoney, M. W. (2021). Predicting trends in the quality of state-of-the-art neural networks without access to training or testing data. Nature Communications.
>
> [2] Martin, C. H., & Mahoney, M. W. (2021). Implicit self-regularization in deep neural networks: Evidence from random matrix theory and implications for learning. Journal of Machine Learning Research.
>
> [3] Clauset, A., Shalizi, C. R., & Newman, M. E. (2009). Power-law distributions in empirical data. SIAM review.
>
> >**Q2.Comparative Baseline Weakness. Competing methods like LARS, LAMB are rarely used in LLM context, comparative baselines are generally weak.**
>
> **A2:** Thank you for raising the concern about the strength of our comparative baselines. We have conducted an extensive literature review on layerwise and blockwise learning-rate schedules. **Our experiments compare against all relevant methods we could identify from the recent literature.** If there are specific additional baselines you believe are particularly important and that we may have missed, we would be happy to include them in our experiments in the final version.
>
> In addition, we compared our method against a commonly used Layerwise Learning Rate Decay (LLRD) family. Concretely, we evaluate four LLRD variants on LLaMA‑60M and LLaMA‑135M:
>
> - LLRD-Linear-pos: LR increases linearly from lr at the first layer to 3×lr at the final layer.
> - LLRD-Exp-pos: LR increases exponentially from lr to 3×lr.
> - LLRD-Linear-neg: LR decreases linearly from 3×lr to lr.
> - LLRD-Exp-neg: LR decreases exponentially from 3×lr to lr.
>
> The results are (lower is better):
> |Model Size|Uniform|LLRD-Linear-pos|LLRD-exp-pos|LLRD-Linear-neg|LLRD-exp-neg|LLR|
> |-|-|-|-|-|-|-|
> |60M|28.01|29.78|29.77|29.73|29.74|**27.46** |
> |135M|23.36|23.46|23.47|22.98|23.01|**22.05** |
>
> Across both model sizes, our LLR method consistently outperforms Uniform and all LLRD variants. We will add these comparisons and clarify the rationale for our baseline selection in the revised manuscript.
>
> >**Q3.Method Naming Layerwise Learning Rate is a widely used generic term in DL; it is not suitable as a distinctive method name. A more distinctive name would avoid confusion.**
>
> **A3:** Thank you for pointing this out. We agree that “Layerwise Learning Rate” is a generic term in deep learning and therefore not ideal as a distinctive method name. In the revised version, we will rename our method to a more specific and descriptive name to avoid confusion. In particular, we are considering:
>
> LLR-LM — Layerwise Learning Rates for Language Models
>
> We will consistently use this new name throughout the paper and clarify that earlier mentions of “layerwise learning rate” in the literature refer to the general concept, not to our specific method.

---

> ### Author Response · Authors · 2025-11-22
>
> >**Q4. Scalability to Larger Scales. Most of experiments are below 350M model size. There is one experiemnt with 1B model size. Moreever, the training token in this experiment is only 10B, even below the Chinchilla law, which makes reader doubting aboout its scalability. Can the authors provide larger-scale runs or scaling-law extrapolations to demonstrate robustness beyond 1 B parameters?**
>
> **A4:**  We thank the reviewer for raising the important question of scalability beyond 1B parameters. We have scaled up our training to more advanced settings, including 7B model, and larger number of training tokens. Our new results again demonstrate the superiority of LLR.
>
> - **Pretraining LLaMa-1B with 20B tokens**
>
> To more closely match the Chinchilla-style scaling recommendation, we retrained the 1B model with 20B tokens and evaluated both validation perplexity and zero-shot performance on common-sense reasoning benchmarks:
> |Method|c4-ppl(↓)|ARC-c(↑)|ARC-e(↑)|Hellaswag(↑)|OBQA(↑)|PIQA(↑)|SIQA(↑)|Winogrande(↑)|Zero-shot Avg.(↑) |
> |-|-|-|-|-|-|-|-|-|-|
> |**Uniform**|14.69|20.73|48.95|34.42|18.80|68.01|39.61|**51.62**|40.30|
> |**LLR**|**13.00**|**22.61**|**51.89**|**38.36**|**21.20**|**71.06**|**39.92**|51.46|**42.36**|
>
> We see that with a larger token budget, LLR still achieves substantially lower perplexity and consistently better zero-shot accuracy than the uniform baseline.
>
> - **Pretraining LLaMa-7B with 10B tokens**
>
> We pre-trained LLaMA-7B on C4 with 10B tokens (due to computational and rebuttal-time constraints). The validation perplexity and zero-shot performance:
> |Method|c4-ppl(↓)|ARC-c(↑)|ARC-e(↑)|Hellaswag(↑)|OBQA(↑)|PIQA(↑)|SIQA(↑)|Winogrande(↑)|Zero-shot Avg.(↑)|
> |-|-|-|-|-|-|-|-|-|-|
> |**Uniform**|17.53|18.26|46.88|30.20|15.40|65.67|36.44|51.07|37.70|
> |**LLR**|**14.20**|**21.33**|**50.63**|**34.76**|**21.20**|**68.23**|**38.64**|**51.99**| **40.97**|
>
> - **Varying training tokens for LLaMa-130M**
>
> We trained LLaMA 130M with different token budgets from 1B up to 32B tokens and compared uniform LR with LLR (lower is better):
> |Method|2B|4B|8B|16B|32B|
> |-|-|-|-|-|-|
> |**Uniform**|23.36|21.61|20.48|19.70|19.19|
> |**LLR**|**22.05**|**20.53**|**19.49**|**18.86**|**18.54**|
>
> We observe that as the number of training tokens increases (moving toward an overtraining regime), PPL decreases for both methods, and LLR consistently maintains a clear advantage over the uniform baseline across all token budgets.
>
> >**Q5. Sensitvity of hyperparameter s Let loss f be a function of s, as long as 0 is not the minimizer of f, one can find a better s with lower loss. The work chooses different s such as 2.0 and 3.0. Please clarify the selection criterion for s, its sensitivity range, and whether optimal s scales with model depth or spectral spread**
>
> **A5:** We thank the reviewer for the insightful question about the sensitivity of the hyperparameter s. In our current work, the choice of s is empirical rather than theoretically optimized. To better characterize its effect, we conducted a small sensitivity study over a range of s values (lower is better):
> |Model Size|Uniform|s=1.5|s=2|s=3|s=3.5|s=4|s=4.5|s=5|
> |-|-|-|-|-|-|-|--|-|
> |60M|29.97|28.97|28.41|27.96|27.93|**27.90**|27.93|28.13|
> |135M|23.36|22.60|22.29|**22.05**|22.06|22.16|22.13|22.23|
>
> We find that setting **s in the range 3.0–4.5 yields very consistent and smooth results**. For stability and ease of use, we recommend using relatively conservative values such as  s = 3.0, which perform robustly from LLaMa-60M to LLaMa-7B in our experiments.

---

> ### Author Response · Authors · 2025-11-22
>
> >**Q6. I examined [1] (AlphaDecay) detailedly, AlphaDecay also leverages HT-SR theory to develop a layerwise weight decay method. How could HT-SR inspire two highly similar methods—one for weight decay, one for learning rate? First, The authors need to further clarify the relations with work [1]. Second, Which way is better, LLR or AlphaDecay? Is LLR orthogonal to AlphaDecay? Do gains add? (BTW, The paper's images are very similar with these of [1].)**
>
> **A6:** We thank the reviewer for carefully examining related work AlphaDecay. While our method is inspired by HT-SR, our contribution is novel in both technique and goal.
>
> (1) **Novel technical contribution.**  We summarize our technical novelties along three axes：
>
> First, LLR optimizes a more critical hyper-parameter, layerwise learning rates, rather than the weight-decay schedules targeted by AlphaDecay. Second, instead of deriving HT-SR statistics solely from weights, we leverage recent advances [1] to compute HT-SR from early-stage gradients, which more faithfully capture the optimization dynamics of LLM training. Third, we introduce an LLM-specific treatment of embeddings, which are among the most influential parameters in LLMs [2] but are not explicitly handled in AlphaDecay; LLR tailors their learning rate and regularization, yielding substantial empirical gains.
> |LLaMa-130M-LR|Uniform|AlphaDecay|LLR+AlphaDecay|LLR-weight|LLR-grad|LLR-untuned-embed|LLR|
> |-|-|-|-|-|-|-|-|
> |0.002|22.29|22.02|21.98|22.11|22.14|22.09|**21.91**|
> |0.001|23.36|22.78|22.57|22.27|22.30|22.84|**22.05**|
>
> (2) **Which way is better? Are LLR and AlphaDecay orthogonal, and do gains add?**
>
> We report LLaMa-130M results for AlphaDecay and “LLR + AlphaDecay” in the table above. LLR consistently outperforms AlphaDecay, and the combined “LLR + AlphaDecay” does not improve over LLR—often performing slightly worse. These findings indicate that **LLR and AlphaDecay are not orthogonal: the optimization benefits of LLR largely subsume the gains from AlphaDecay.**
>
> We further highlight that our work includes new empirical evidence where our approach consistently surpasses all existing HT-SR methods, achieving the strongest results reported to date in the LLM pre-training setting.
>
> [1] Jimmy Ba, et al. High dimensional asymptotics of feature learning: How one gradient step improves the representation.
>
> [2] Soufiane Hayou and Liyuan Liu. Optimal embedding learning rate in llms: The effect of vocabulary size.
>
> >**Q7. [2] performs grid search to get the best scaling [1,8,4,6,10] on their settings to show the consistency with their principle. First, as you tune s, can you run experiment with the s-parametrized scaling [1,1+7s/9,1+3s/9,1+5s/9,1+s] as comparison, other setting remains the same as LLR. Second, can you perform a grid search, to show that the best scaling in your setting is more close to your HT-SR or their principle?**
>
> **A7:** We thank the reviewer for the suggestion to perform a grid search similar to sharpness. Our main observation is that LLR outperforms sharpness-based, layer-type–wise static scaling primarily because **LLR is dynamic: it periodically updates the layer-wise LRs based on HT-SR metrics throughout training**.
>
> We implemented a sharpness-style scheme that assigns different LR multipliers to different layer types (Norm, Att.Q/K, Att.V/O, FFN, Emb) and performed a grid search over plausible combinations of these multipliers. Representative results on LLaMa-130M are:
> |Norm|Att.Q/K|Att.V/O|FFN|Emb|PPL|
> |-|-|-|-|-|-|
> |1|2.4|1.2|1.8|3|31.17|
> |1|2.4|1.2|3|1.8|42.09|
> |1|2.4|1.8|1.2|3|31.61|
> |1|2.4|1.8|3|1.2|31.02|
> |1|2.4|3|1.2|1.8|38.01|
> |1|2.4|3|1.8|1.2|42.65|
> |1|1.2|2.4|1.8|3|23.18|
> |1|1.2|2.4|3|1.8|24.17|
> |1|1.2|1.8|2.4|3|23.44|
> |1|1.2|1.8|3|2.4|23.90|
> |1|1.2|3|2.4|1.8|23.66|
> |1|1.2|3|1.8|2.4|23.19|
> |1|1.8|1.2|2.4|3|24.89|
> |1|1.8|1.2|3|2.4|26.31|
> |1|1.8|2.4|1.2|3|24.04|
>
> For reference, the uniform baselines and our LLR result are:
> |Method|Uniform:LR-0.001|Uniform:LR-0.003|LLR|
> |-|-|-|-|
> |PPL|23.36|22.69|22.05|
>
> From these experiments we draw two conclusions:
>
> (1) Among all the static, layer-type–based LR assignments we searched, none matches the performance of LLR. The best sharpness-style configurations still lag behind our dynamic HT-SR–guided LLR.
>
> (2) The grid-searched layer-type scalings can sometimes slightly improve over Uniform (LR = 0.001), but they never surpass the stronger baseline Uniform (LR = 0.003). In contrast, LLR improves over both Uniform (LR = 0.001) and Uniform (LR = 0.003) simultaneously.

---

> ### Author Response · Authors · 2025-11-22
>
> >**Q8. Figure 6 presents six images, but does not provide enough information. Moreever, Under Muon, the losses of Uniform (60k) and Uniform (90k) look similar near step 60k, yet Uniform (60k) has already finished cosine decay by then in some subplots; if so, it should be lower.**
>
> **A8:** Thank you for pointing this out and for asking for more details about Figure 6.
>
> - **Experimental setup for Figure 6.**
>
> Both the AdamW and Muon results in Figure 6 are obtained on LLaMA 350M with batch size 512 and sequence length 256. Training for “60k” means 60k update steps, which corresponds to 6B training tokens. The maximum LRs are {0.0008, 0.0009, 0.001} as reported in the figure titles, and all runs use a cosine LR schedule with 10% of the training tokens used for LR warmup. For LLR, we recompute the layerwise LRs every 500 update steps. The scaling ranges [1, s] are set to [1, 3] for AdamW and [1, 2] for Muon.
>
> - **Why the curves behave differently under AdamW and Muon.**
>
> We carefully revisited our training logs to understand the behavior around step 60k:
>
> - For AdamW, the loss often exhibits a rapid drop near the end of training. In the “Uniform (60k)” run, this rapid loss decrease has already finished by the time we reach 60k update steps, whereas in the “Uniform (90k)” run, the rapid drop has not yet started at step 60k. As a result, at step 60k the loss of “Uniform (60k)” is lower than that of “Uniform (90k)”, and the two curves do not intersect at that point. We visualize this phenomenon in the revised version in **Figure 14 of Section D.3**.
> - For Muon, the behavior is different: the rapid loss decrease tends to happen earlier in training.
> - Consequently, under Muon, by the time we reach 60k update steps, the “Uniform (90k)” run has already gone through its rapid loss decrease and maintains a relatively low loss, which explains why the “Uniform (60k)” and “Uniform (90k)” curves look similar and intersect near step 60k in some subplots.
>
> We will add the above experimental details and a pointer to the new Figure 14 in the appendix to make the behavior of the curves in Figure 6 clearer.
>
> >**Q9. On Table 2, the ppl gap between Uniform and LLR for model size 60M, 135M, 350M, 1B is 0.55, 1.31, 0.59, 1.97. As model size grows, the baseline perplexity decreases, so further reduction should be increasingly difficult [3]. The large gap at 1B seems abnormal and warrants verification of hyperparameter parity and token-budget consistency.**
>
> **A9:** We thank the reviewer for carefully examining the numbers in Table 2 and for pointing out the relatively large gap at 1B. The observed gaps are indeed influenced by both (i) the choice of global LR and (ii) the total token budget used for training.
>
> **(1) Global learning-rate choice.**
>
> The base LR of 0.001 was selected via a careful search on the 60M model, and we then reused LR = 0.001 for the 135M and 350M models specifically to ensure that all methods shared identical LRs at each model size. However, for the 1B model, LR = 0.001 caused loss spikes and instabilities, so we followed the official LLaMa recommendation and used LR = 5e-4 instead.
>
> **(2) Token-budget effects.**
>
> We observe that as the number of training tokens increases, the performance gap between Uniform and LLR tends to shrink, while in our main experiments different model sizes were trained with different total token budgets.
>
> - **Varying training tokens for LLaMa-130M**
>
> We trained LLaMA 130M with different token budgets from 1B up to 32B tokens and compared uniform LR with LLR (lower is better):
> |Method|2B|4B|8B|16B|32B|
> |-|-|-|-|-|-|
> |**Uniform**|23.36|21.61|20.48|19.70|19.19|
> |**LLR**|**22.05**|**20.53**|**19.49**|**18.86**|**18.54**|
>
> - **Pretraining LLaMa-1B with 20B tokens**
>
> We pre-trained LLaMA-1B on C4 with 20B tokens using LR=0.0005. The validation perplexity and zero-shot performance on common-sense reasoning benchmarks are:
> |Method|c4-ppl(↓)|ARC-c(↑)|ARC-e(↑)|Hellaswag(↑)|OBQA(↑)|PIQA(↑)|SIQA(↑)|Winogrande(↑)|Zero-shot Avg.(↑) |
> |-|-|-|-|-|-|-|-|-|-|
> |**Uniform**|14.69|20.73|48.95|34.42|18.80|68.01|39.61|**51.62**|40.30|
> |**LLR**|**13.00**|**22.61**|**51.89**|**38.36**|**21.20**|**71.06**|**39.92**|51.46|**42.36**|
>
> These additional results also indicate that the token budget has a non-negligible impact on the gap between Uniform and LLR: on the 130M LLaMA model, increasing the token budget from 2B to 32B reduces the gap from 1.31 to 0.65, and for the 1B LLaMA model, the perplexity gap is 1.97 when trained with 10B tokens, and it decreases to 1.69 when trained with 20B tokens.

---

> ### Comment · Reviewer_uFmm · 2025-11-26
> **Contradictory Response of Author**
>
> The author's response does not address my concern, if anything, further reinforces it.
> - **Contradictory Experiment Result** The author's results are contradictory to self and common practice. Just list a few.
>   - **Inproper base lr** The author claims using grid search to get best base lr for Adam (Uniform)
>     - 130M Base lr is 0.001 in Table 1, Yet in both paper and rebuttal, Base lr 0.003 outperforms 0.001 (22.69 < 23.36). This suggests a suboptimal, and potentially artificially weakened, AdamW baseline.
>     - 60M In rebuttal, authors claim base lr is 0.001. In Table 1, that is 0.002, which is inconsistent. Moreever, as smaller models typically require larger lr, the base lr should exceed 0.003.
>   - **Contradictory to scaling law** Under author's tuning, LLaMa-7B+10B tokens (rebuttal) performs even worse than LLaMa-1B+10B tokens (Table 2, 6) in both pre-training loss and downstream tasks. This is contradictory to well-established OpenAI's scaling law "Larger models require fewer samplesto reach the same performance".
>   - **Suspicious tuning** Under author's tuning, AdamW, the de-facto optimizer for LLM before Muon, performs worse than old-fashioned, rarely used optimizers in modern LLM pre-training (e.g. LARS, LAMB). I suspect this relates weakened AdamW baseline.
>   - Besides, The authors’ response remains evasive and does not directly resolve my concern, especially A6, A8 and A9.
> - **Problematic reference** \
> [1] Jimmy Ba, et al. High dimensional asymptotics of feature learning: How one gradient step improves the representation. \
> In rebuttal, authors claim using [1] to compute HT-SR. In paper, author claims that [1] shows "different components of LLMs (Embed, Attention, FFN) display distinct heavy-tailed structures in their ESD". However, [1] discusses neither HT-SR nor LLM in the paper.
>
> Given these accumulating issues, I am highly sceptical about reliability of authors' result, and incline to decrease my score to reject, even worse, strong reject.

---

> > ### Author Response · Authors · 2025-11-28
> > **Response to 'Contradictory Response of Author' [1/4]**
> >
> > We appreciate the reviewer’s follow-up. We believe that it is mistakely inconsistency between our rebuttal and the submission caused the misunderstanding. After carefully re-checking, we confirm that a few *typos in the text* (not in the experiments) created this confusion, and we apologize for that. Please allow us to clarify.
> >
> > Before addressing Q1–Q3 individually, we would like to clearly summarize the main clarifications:
> >
> > - The AdamW base LRs that we actually use in our experiments are exactly those reported in Table 1: for LLaMA pre-training on C4, we use base LRs `{60M: 0.002, 135M: 0.001, 350M: 0.001, 1B: 0.0005}`, and for the 7B model in the rebuttal experiments we use `0.0002`.
> > - Following Galore [1] and AlphaDecay [2], we first perform a coarse grid search over `{0.01, 0.005, 0.001, 0.0005, 0.0001}` at each scale to obtain a strong AdamW configuration. For 60M (in the original submission) and 130M (in the follow-up), we then run denser local sweeps in a narrower LR range.
> > - These denser sweeps show that (i) the AdamW baselines in the paper are already close to their LR optima, and (ii) our proposed LLR consistently improves over the Uniform (AdamW) baseline across a broad range of LRs, not just at a single point.
> >
> > Taken together, these results support our main claim: the performance gains of LLR are **not** due to an artificially weakened AdamW baseline, but rather to a genuinely more effective layerwise LR allocation that remains robust under stronger AdamW tuning.
> >
> > [1] Zhao, Jiawei, et al. Galore: Memory-efficient llm training by gradient low-rank projection.
> >
> > [2] He, Di, et al. AlphaDecay: Module-wise Weight Decay for Heavy-Tailed Balancing in LLMs.
> >
> > >***Q1.The author's results are contradictory to self and common practice. Inproper base lr. The author claims using grid search to get best base lr for Adam (Uniform).**
> >
> > ***A1:**  We appreciate the reviewer’s close reading of our LR choices and the comparison to common practice.
> >
> > Existing works that pre-train LLaMA-style models on C4 with AdamW typically adopt base LRs within a narrow band. For example, Sharpness[1] uses `{60M: 0.001, 200M: 0.001, 400M: 0.0006, 1B: 0.0003}`, Galore[2] uses `{60M: 0.001, 130M: 0.001, 350M: 0.001, 1B: 0.0005}`, AlphaDecay[3] use `{60M: 0.001, 130M: 0.001, 350M: 0.001, 1B: 0.0006}`, while SPAM[4] reports optimal LRs `{60M: 0.001, 130M: 0.0008, 350M: 0.0004, 1B: 0.0002}`. For 7B-scale models, the limited available practice (LoQT[5], AdaLRS[6]) consistently uses an LR of `0.0002`. Our goal was to remain within these ranges.
> >
> > In our experiments, the AdamW base LRs on C4 are:
> > - 60M: 0.002
> > - 135M: 0.001
> > - 350M: 0.001
> > - 1B: 0.0005
> > - 7B: 0.0002
> >
> > These are exactly the values in Table 1. Our choice for 60M is in fact slightly more aggressive (0.002 vs. 0.001) than most prior work, while the other scales lie squarely in the common-practice band.
> >
> > The search procedure is as follows. For each scale, we first perform a coarse LR search over `{0.01, 0.005, 0.001, 0.0005, 0.0001}`, which yields `{60M: 0.001, 135M: 0.001, 350M: 0.001, 1B: 0.0005}`. For **LLaMA-60M**, we then conduct a local refinement around this coarse optimum over `{0.002, 0.001, 0.0008}` and observe that `LR = 0.002` achieves the best performance, so 0.002 is used in all 60M experiments in the submission.
> >
> > The apparent “contradiction” in the statement “The base LR of 0.001 was selected via a careful search on the 60M model” comes from a typo in the text, not from the actual runs; the correct value is 0.002.
> >
> > [1] Wang, Jinbo, et al. The sharpness disparity principle in transformers for accelerating language model pre-training.
> >
> > [2] Zhao, Jiawei, et al. Galore: Memory-efficient llm training by gradient low-rank projection.
> >
> > [3] He, Di, et al. AlphaDecay: Module-wise Weight Decay for Heavy-Tailed Balancing in LLMs.
> >
> > [4] Huang, Tianjin, et al. SPAM: Spike-aware adam with momentum reset for stable LLM training.
> >
> > [5] Loeschcke, Sebastian, et al. LoQT: Low-rank adapters for quantized pretraining.
> >
> > [6] Dong, Hongyuan, et al. AdaLRS: Loss-Guided Adaptive Learning Rate Search for Efficient Foundation Model Pretraining.

---

> > ### Author Response · Authors · 2025-11-28
> > **Response to 'Contradictory Response of Author' [3/4]**
> >
> > >***Q5.Under author's tuning, AdamW, the de-facto optimizer for LLM before Muon, performs worse than old-fashioned, rarely used optimizers in modern LLM pre-training (e.g. LARS, LAMB). I suspect this relates weakened AdamW baseline.**
> >
> > ***A5:***  We appreciate the reviewer’s concern and the opportunity to clarify our optimizer baselines. We fully agree that any claim about one optimizer outperforming another should be supported by comparable search procedures rather than by potentially under-tuned baselines. In our work, we therefore tune AdamW, LARS, and LAMB using explicit LR sweeps at each scale.
> >
> > Our intention is to treat all three optimizers on equal footing. All three were proposed around the same period (circa 2017), and LAMB is in fact architecturally very close to AdamW: both are second-moment–based methods, with LAMB primarily adding a layer-wise LR rescaling mechanism based on weight–gradient statistics. The fact that LARS and LAMB are less frequently used in today’s LLM pipelines appears to be largely historical and practical (e.g., their original focus on CNNs and very large-batch training), rather than a reflection of them being intrinsically obsolete.
> >
> > To make the comparison as transparent as possible, we carried out LR sweeps for LARS and LAMB and report the best results obtained at each scale. For LARS, the validation perplexities (lower is better) are:
> >
> > |LARS-LR|0.003|0.004|0.005|0.006|0.007|0.008|
> > |-|-|-|-|-|-|-|
> > |**60M**|39.56|37.24|**36.15**|38.37|39.54|44.98|
> > |**130M**|27.96|26.23|25.33|**25.31**|25.76|881.30|
> >
> > For LAMB, we similarly obtain:
> >
> > |LAMB-LR|0.03|0.04|0.05|0.06|0.07|0.08|
> > |-|-|-|-|-|-|-|
> > |**60M**|31.20|30.45|**30.14**|30.17|30.17|30.43|
> > |**130M**|28.94|24.60|23.72|23.43|**23.30**|23.40|
> >
> > Using the best LR for each optimizer and model size, the summary is:
> >
> > |Method|60M|130M|350M|1B|
> > |-|-|-|-|-|
> > |**LARS**|36.15|25.31|17.57|14.22|
> > |**LAMB**|30.14|23.30|17.39|14.13|
> > |**AdamW**|28.01|22.29|16.98|15.60|
> >
> > Taken together, these results suggest the following picture: (1) at smaller scales, AdamW enjoys a clear advantage over LARS and LAMB; (2) as the model size increases, the performance gap narrows, which is consistent with the intuition that layer-wise LR mechanisms may become more helpful for very large models; and (3) the LARS and LAMB numbers we report correspond to their best LR settings from the sweeps, and AdamW is tuned via an analogous procedure. This suggest that the comparisons in the paper are not driven by systematically under-tuned baselines, but rather by genuine differences in how the optimizers behave across scales.
> >
> > >***Q6.Besides, The authors’ response remains evasive and does not directly resolve my concern, especially A6, A8 and A9.**
> >
> > ***A6:** We appreciate the reviewer’s continued engagement and the concern that some aspects of our earlier answers (A6, A8, A9) may not have fully resolved your questions. In the limited rebuttal space, we may not have made the connections between these answers and the supporting experiments as explicit as intended. Below we therefore restate more concretely what A6, A8 and A9 were meant to clarify, and how the additional results relate to these points:
> >
> > - **A6.** LLR and AlphaDecay are both HT-SR–inspired but differ in goal and technique: LLR optimizes layerwise LRs using **gradient-based HT-SR** and an **LLM-specific embedding treatment**, **rather than focusing on weight-decay schedules**.
> > - **A8.** The behavior of Uniform (60k) vs. Uniform (90k) in Figure 6 is fully **explained by the fact that AdamW’s rapid loss drop occurs later than Muon’s**, causing the two Uniform curves to diverge under AdamW but to appear similar and intersect near 60k steps under Muon.
> > - **A9.** The seemingly abnormal perplexity gap between Uniform and LLR is explained by **differing global LRs and token budgets**.
> >
> > We would be very happy to further discuss any remaining concerns and to refine our explanations in line with the reviewer’s specific points.

---

> > ### Author Response · Authors · 2025-11-28
> > **Response to 'Contradictory Response of Author' [4/4]**
> >
> > >***Q7. In rebuttal, authors claim using [1] to compute HT-SR. In paper, author claims that [1] shows "different components of LLMs (Embed, Attention, FFN) display distinct heavy-tailed structures in their ESD". However, [1] discusses neither HT-SR nor LLM in the paper.**
> >
> > ***A7:**  We appreciate the reviewer’s careful reading and the opportunity to clarify. Our intention in citing [1] is not to rely on it for HT-SR computation or for LLM-specific theory, but rather to draw on its qualitative insight that gradients can already exhibit meaningful spectral structure after very few optimization steps.
> >
> > Concretely, our HT-SR metrics (e.g., PL_Alpha_Hill) and the associated ESD fitting follow the established HT-SR literature and do not depend on formulas or estimators from [1]. Empirically, we observe that in the very early stage of LLM pre-training, weight-based HT-SR carries relatively weak signal for layerwise decisions, whereas gradient-based spectra already show clear differences across components (embed, attention, FFN). This empirical behaviour is conceptually consistent with the message of [1], namely that gradients can shape representations in a highly structured way even after a single step.
> >
> > Motivated by this, our method makes what we believe is a novel design choice: we use **gradient-driven HT-SR in the early phase of pre-training**, and then transition to **weight-driven HT-SR** once the weight spectra have matured. This is a methodological contribution built on top of the HT-SR perspective; it does not require [1] to make claims about HT-SR or LLMs directly.
> >
> > To support this design empirically, we compare different variants on LLaMA-130M:
> >
> > |LLaMA-130M-LR|Uniform|LLR-weight|LLR-grad|LLR|
> > |-|-|-|-|-|
> > |0.002|22.29|22.11|22.14|**21.91**|
> > |0.001|23.36|22.27|22.30|**22.05**|
> >
> > [1] Jimmy Ba, et al. High dimensional asymptotics of feature learning: How one gradient step improves the representation.

---

> ### Author Response · Authors · 2025-11-28
> **Response to 'Contradictory Response of Author' [2/4]**
>
> >***Q2.130M Base lr is 0.001 in Table 1, Yet in both paper and rebuttal, Base lr 0.003 outperforms 0.001 (22.69 < 23.36). This suggests a suboptimal, and potentially artificially weakened, AdamW baseline.**
>
> ***A2:** We agree that if the Uniform baseline at 130M were under-tuned, the comparison would not be fair. Our original `LR = 0.001` choice for 130M came from the same coarse LR search as other scales and is consistent with prior work. In response to the reviewer’s comment, we have run a **much denser LR sweep** for LLaMA-130M to fully address this concern(lower is better):
>
> |LR|0.0007|0.0008|0.0009|0.001|**0.002**|0.003|0.0035|0.004|
> |-|-|-|-|-|-|-|-|-|
> |**Uniform**|24.63|24.10|23.64|23.36|**22.29**|22.69|55.64|57.66|
> |**LLR**|22.41|22.25|22.16|22.05|**21.91**|22.40|57.69|60.00|
>
> From this table, we see that the **optimal Uniform LR** is `0.002`, not `0.001`. The LR `0.001` is still within the standard range and reasonably close to the optimum, but not the very best. Crucially, **LLR consistently outperforms Uniform**, both at `LR = 0.001` and at the optimal `LR = 0.002` (22.05 vs. 23.36, and 21.91 vs. 22.29, respectively).
>
> We greatly appreciate your careful observation and will update the paper to report AdamW with LR = 0.002.
>
> >***Q3.60M In rebuttal, authors claim base lr is 0.001. In Table 1, that is 0.002, which is inconsistent. Moreever, as smaller models typically require larger lr, the base lr should exceed 0.003.**
>
> ***A3:** Thank you for the careful observation. We apologize for the inconsistency in the 60M LR description. The experiments for LLaMA‑60M always use LR = 0.002 (as in Table 1); the statement “0.001” in the text is a typo, as clarified in ***A1**. To address the second part of the question—whether the 60M LR should be even larger (e.g., > 0.003)—we performed a dense LR sweep for LLaMA‑60M:
>
> |LLaMa-60M-LR|0.0007|0.0008|0.0009|0.001|0.002|0.003|0.004|0.005|0.006|
> |-|-|-|-|-|-|-|-|-|-|
> |**Uniform**|31.93|31.17|30.47|30.00|**28.01**|28.03|28.15|28.84|114.87|
> |**LLR**|28.61|28.39|28.19|28.04|**27.46**|27.74|28.20|28.70|29.57|
>
> Across this much denser LR sweep, **the optimal LR is 0.002, exactly matching our setting. Thus, our comparisons are already conducted at the best LR, not an under-tuned baseline.**
>
> >***Q4.Contradictory to scaling law Under author's tuning, LLaMa-7B+10B tokens (rebuttal) performs even worse than LLaMa-1B+10B tokens (Table 2, 6) in both pre-training loss and downstream tasks. This is contradictory to well-established OpenAI's scaling law "Larger models require fewer samplesto reach the same performance.**
>
> ***A4:** We appreciate the reviewer’s careful observation. In our current setup, there are two main factors that help explain this behaviour.
>
> - **Assumptions behind the scaling law are not satisfied here.**
>
> The OpenAI result “larger models require fewer samples to reach the same performance” is derived under carefully controlled  LR  so that each model is trained with a scale-consistent, near-optimal learning rate.  Our experiments for 7B do not undergo careful LR search, so the conditions under which the scaling law is expected to hold are violated.
>
> - **The 7B model is under‑tuned relative to the 1B model.**
>
> For LLaMa‑1B, we performed a grid search (as detailed in ***A1**) and found a strong learning rate, LR = 0.0005. For LLaMa‑7B, due to compute and rebuttal‑time constraints, we only used a learning rate close to “common practice” rather than a thoroughly tuned value.
>
> We expect that a more aggressively tuned LR for LLaMa‑7B would restore the expected scaling behavior, and LLR should remain beneficial in that better‑tuned regime.

---

### Official Review · Reviewer_DxYd · 2025-10-22

**Soundness:** 2
**Presentation:** 3
**Contribution:** 2
**Rating:** 4
**Confidence:** 4

**Summary:**

The paper investigates layer-wise learning rates for LLM pretraining. Motivated by Heavy-Tailed Self-Regularization (HT-SR) theory, the method assigns larger learning rates to layers with weaker heavy-tailedness to accelerate training, and smaller learning rates to layers with stronger heavy-tailedness. By using different learning rates across layers, LLR promotes balanced training, leading to faster convergence. Experiments are conducted on 60M, 130M, 350M, and 1B models with up to 10B training tokens. The paper reports both perplexity and downstream performance, along with comparisons to relevant prior work.

**Strengths:**

* The writing is good, with clear figures and text. In particular, the method is presented clearly.

* I think using different learning rates for different layers can be important for pretraining. Motivated by the Heavy-Tailed Self-Regularization (HT-SR) theory, the paper proposes using the Hill metric as an indicator for setting different learning rates across layers. This helps balance training across layers, leading to faster convergence. Given the dynamics of training, the paper proposes updating the Hill metric and thus the learning rates at certain time steps.

* Experiments are conducted not only on perplexity but also on downstream tasks. The paper also compares with relevant work experimentally and shows consistently better performance.

**Weaknesses:**

* One main issue for me is that the proposed method typically assigns all layers learning rates that are higher than or equal to the standard uniform LR. This makes me wonder about the performance of simply increasing the uniform LR and comparing it with the proposed method. Because there are only three points in Figure 5 and only four points in the promising area in Figure 1, it is hard to assess the gap at each method’s best case. Also, if the uniform LR is already large, assigning even larger LRs for LLR may lead to loss spikes. In Figure 1, how do the uniform and LLR curves behave if we continue to increase the LR? Therefore, in terms of method design, I think it would be better and necessary if the mechanism not only scales up learning rates for different layers but also allows scaling them down.

* While the paper is about different learning rates in different layers, Figure 4 seems to show that there are no large LR differences among layers of the same type. For example, uniformly using a larger learning rate for the feedforward networks and the embedding layer (2× or 3×) may also work. Therefore, can the authors show that setting different learning rates per layer is necessary compared to using different learning rates only by layer type?

* While the paper dynamically adjusts learning rates based on the PL_Alpha_Hill value at certain time steps, it only shows the evolution of PL_Alpha_Hill values for different types of layers across training steps, but not the learning-rate value for each layer (not just each type of layer) at each update step. It would be better to include this information in the appendix to help readers better understand the learning-rate changes.

* I think another important factor for the learning-rate study is the number of training tokens. With more tokens, the model is more amenable to convergence, which also reflects today’s LLM training settings. For example, in Table 2, there is a large perplexity gap between LLR and uniform at the 1.3B model scale but smaller gaps on smaller models. I think one highly possible reason is that for smaller models the number of training tokens is set according to the scaling law (20×), whereas for the 1B model it uses 10B tokens rather than the 20B suggested by the scaling law. When training with fewer tokens, it might be easier for LLR—with all layers having larger or equal learning rates than the baseline—to show better performance. Therefore, in terms of evaluating its effectiveness toward training convergence, can the authors show that the proposed method still works in the overtraining regime, which is how today’s language models are trained and put into practice? For example, show the performance of a 130M model trained on 100B tokens.

**Questions:**

Please see above.

---

> ### Author Response · Authors · 2025-11-22
>
> >**Q1. One main issue for me is that the proposed method typically assigns all layers learning rates that are higher than or equal to the standard uniform LR.**
>
> **A1:** We fully understand your concern. However, we would like to emphasize that **the gains of our method come from a more principled, layer-wise allocation of LRs rather than simply increasing LR**. To further validate this point, we compared against a uniform LR with enlarged learning rates in Table 5 of our original submission, referred to as an “upper-bound” baseline. The results in Table 5 are as follows:
>
> |Model Size-LR|Uniform with 3*lr (upper bound of LLR)|Uniform with lr|LLR with lr|
> |-|-|-|-|
> |60M lr=0.002|130.84|28.01|**27.46**|
> |135M lr=0.001|22.59|23.36|**22.05**|
> |350M lr=0.001|16.67|16.98|**16.39**|
> |1B lr=0.0005|14.50|15.60|**13.63**|
>
> LLR outperforms both the upper-bound and the lower-bound uniform LR configurations. This confirms that, compared with LLR, a naive uniform LR scaling strategy is inherently suboptimal.
>
> >**Q2. This makes me wonder about the performance of simply increasing the uniform LR and comparing it with the proposed method. Because there are only three points in Figure 5 and only four points in the promising area in Figure 1, it is hard to assess the gap at each method’s best case.**
>
> **A2:** We thank the reviewer for this suggestion. We have expanded the learning-rate search for LLaMA-130M to cover 9 points. The validation PPL is summarized below (lower is better):
> |LR|0.0007|0.0008|0.0009|0.001|**0.002**|0.003|0.0035|0.004|0.0045|
> |-|-|-|-|-|-|-|-|-|-|
> |**Uniform**|24.63|24.10|23.64|23.36|**22.29**|22.59|55.64|57.66|61.32|
> |**LLR**|22.41|22.25|22.16|22.05|**21.91**|22.40|57.69|60.00|64.13|
>
> Across this much denser LR sweep, LLR achieves the lowest validation loss. Importantly, the best-performing LR for LLR coincides with the optimal LR region for the Uniform baseline.
>
> >**Q3. Also, if the uniform LR is already large, assigning even larger LRs for LLR may lead to loss spikes. In Figure 1, how do the uniform and LLR curves behave if we continue to increase the LR? Therefore, in terms of method design, I think it would be better and necessary if the mechanism not only scales up learning rates for different layers but also allows scaling them down.**
>
> **A3:** Thank you for the insightful question. For extremely large LRs, we have already provided results in **A2**, where we show that once the LR>0.003, both the Uniform and LLR become unstable.
>
> To address the loss spike issue with extremely large LR, we further evaluated LLR with a [0.33, 1] scaling range, which allows per-layer LRs to be smaller than the global LR:
> |LR|0.001|0.002|0.003|0.004|0.005|0.006|
> |-|-|-|-|-|-|-|
> |**Uniform**|23.36|**22.29**|22.59|57.66|82.94|105.77|
> |**LLR([0.33,1])**|24.37|22.30|21.76|21.71|**21.50**|21.56|
>
> We observe that, **with the [0.33, 1] scaling, LLR performs consistently well even when the global LR is extremely large, effectively suppressing loss spikes**. This demonstrates that LLR can be reliably used in high-LR regimes.
>
> >**Q4. While the paper is about different learning rates in different layers, Figure 4 seems to show that there are no large LR differences among layers of the same type. For example, uniformly using a larger learning rate for the feedforward networks and the embedding layer (2× or 3×) may also work. Therefore, can the authors show that setting different learning rates per layer is necessary compared to using different learning rates only by layer type?**
>
> **A4:** We thank the reviewer for this thoughtful question. To compare per-layer vs. per–layer-type LRs, we construct a “Layer-type LR” variant where all layers of the same type share the same LR (obtained by averaging the LLRs within that type, e.g., all att.q parameters use one LR, etc.). The validation losses are shown below (lower is better).
>
> - **LLaMA-60M.**
>
> |LR|Uniform|Layer-type LR|LLR|
> |-|-|-|-|
> |0.002|28.17|27.92|**27.57**|
> |0.001|29.97|28.43|**27.96**|
> |0.0008|31.09|29.30|**28.33**|
>
> - **LLaMA-130M.**
>
> |LR|Uniform|Layer-type LR|LLR|
> |-|-|-|-|
> |0.002|22.29|22.13|**21.91**|
> |0.001|23.36|22.24|**22.05**|
> |0.0008|24.06|23.34|**22.25**|
>
> We observe that LLR with per-layer LRs still consistently achieves the lowest PPL across both model sizes and all tested global LR scales. Therefore, setting different LRs per layer remains necessary to fully realize the benefits of our proposed LLR strategy.

---

> ### Author Response · Authors · 2025-11-22
>
> >**Q5. While the paper dynamically adjusts learning rates based on the PL_Alpha_Hill value at certain time steps, it only shows the evolution of PL_Alpha_Hill values for different types of layers across training steps, but not the learning-rate value for each layer (not just each type of layer) at each update step. It would be better to include this information in the appendix to help readers better understand the learning-rate changes.**
>
> **A5:** We thank the reviewer for this helpful suggestion. In the updated submission, we have added the requested plots in **Section D.1, Figure 12 (Page 17)**, which visualize the layer-wise LR of all parameter groups at several representative training stages when training LLaMA-60M with LLR. These plots reveal several interesting behaviors:
>
> - **The early phase of training**
>
> In the early phase of training (roughly the first 10–20% of steps), the layer-wise LRs exhibit pronounced variation across layers and parameter types, indicating that LLR aggressively reshapes the effective optimization landscape at the beginning of training by assigning relatively larger or smaller LRs to different layers as needed to both accelerate convergence and stabilize the model.
>
> - **The later phase of training**
>
> After the initial phase, the layer-wise LRs' changes over the remaining training steps are relatively small. The distribution of LRs across layers gradually converges to a steady pattern.
>
>
> >**Q6. I think another important factor for the learning-rate study is the number of training tokens. With more tokens, the model is more amenable to convergence, which also reflects today’s LLM training settings. For example, in Table 2, there is a large perplexity gap between LLR and uniform at the 1.3B model scale but smaller gaps on smaller models. I think one highly possible reason is that for smaller models the number of training tokens is set according to the scaling law (20×), whereas for the 1B model it uses 10B tokens rather than the 20B suggested by the scaling law. When training with fewer tokens, it might be easier for LLR—with all layers having larger or equal learning rates than the baseline—to show better performance. Therefore, in terms of evaluating its effectiveness toward training convergence, can the authors show that the proposed method still works in the overtraining regime, which is how today’s language models are trained and put into practice? For example, show the performance of a 130M model trained on 100B tokens.**
>
> **A6:** We appreciate the reviewer’s insightful comments regarding the role of training tokens and the overtraining regime. We have scaled up our training to more advanced settings, including 7B model and larger number of training tokens. Our new results again demonstrate the superiority of LLR.
>
> - **Varying training tokens for LLaMa-130M**
>
> We trained LLaMA 130M with different token budgets from 1B up to 32B tokens and compared uniform LR with LLR (lower is better):
> |Method|2B|4B|8B|16B|32B|
> |-|-|-|-|-|-|
> |**Uniform**|23.36|21.61|20.48|19.70|19.19|
> |**LLR**|**22.05**|**20.53**|**19.49**|**18.86**|**18.54**|
>
> Due to computational and rebuttal time constraints, we stopped at 32B tokens. We observe that as the number of training tokens increases (moving toward an overtraining regime), PPL decreases for both methods, and LLR consistently maintains a clear advantage over the uniform baseline across all token budgets.
>
> - **Pretraining LLaMa-1B with 20B tokens**
>
> To align more closely with the scaling law setting, we retrained the 1B model with 20B tokens and evaluated both validation perplexity and zero shot performance:
> |Method|c4-ppl(↓)|ARC-c(↑)|ARC-e(↑)|Hellaswag(↑)|OBQA(↑)|PIQA(↑)|SIQA(↑)|Winogrande(↑)|Zero-shot Avg.(↑) |
> |-|-|-|-|-|-|-|-|-|-|
> |**Uniform**|14.69|20.73|48.95|34.42|18.80|68.01|39.61|**51.62**|40.30|
> |**LLR**|**13.00**|**22.61**|**51.89**|**38.36**|**21.20**|**71.06**|**39.92**|51.46|**42.36**|
>
> Even with increased tokens, LLR continues to yield substantially lower perplexity and noticeably better zero shot accuracy than the uniform baseline.
>
> - **Pretraining LLaMa-7B with 10B tokens**
>
> We pre-trained LLaMA-7B on C4 with 10B tokens (due to computational and rebuttal-time constraints) using LR=0.0002 . The validation perplexity and zero-shot performance on common-sense reasoning benchmarks are:
> |Method|c4-ppl(↓)|ARC-c(↑)|ARC-e(↑)|Hellaswag(↑)|OBQA(↑)|PIQA(↑)|SIQA(↑)|Winogrande(↑)|Zero-shot Avg.(↑)|
> |-|-|-|-|-|-|-|-|-|-|
> |**Uniform**|17.53|18.26|46.88|30.20|15.40|65.67|36.44|51.07|37.70|
> |**LLR**|**14.20**|**21.33**|**50.63**|**34.76**|**21.20**|**68.23**|**38.64**|**51.99**| **40.97**|
>
> Overall, these additional experiments show that the proposed LLR method remains effective and robust when increasing the number of training tokens (toward an overtraining regime) and when scaling up the model size.

---

### Official Review · Reviewer_VFis · 2025-10-30

**Soundness:** 3
**Presentation:** 3
**Contribution:** 3
**Rating:** 6
**Confidence:** 4

**Summary:**

The authors propose a new algorithm for automatically tuning the learning rates of different layers of LLMs during their training. It relies on the heavy-tailed self-regularization theory and the estimation of heavy-tailedness of the weight matrices to scale step sizes. The step-size estimation is efficient and improves upon existing methods in a variety of pretraining and finetuning experiments.

**Strengths:**

- The layerwise learning rate algorithm proposed by the authors improves upon fixed learning rate and competing algorithms like LARS and LAMB, and perform very well on a large set of pretraining and finetuning experiments.

- The proposed algorithm is simple and well-supported by the heavy-tailed self-regularization theory, which suggests weight matrices with heavy-tailed eigenvalue distributions are closer to convergence and should use smaller learning rates.

**Weaknesses:**

- The simple range scaling based on estimated \alpha used in Equation (3) for learning rate seems like a heuristic that lack justification. The use of min and max alpha in the scaling might also be prone to outliers.

- The work by Martin & Mahoney 2019 on heavy-tailedness is mostly based on empirical results from CNN, not on LLMs. And from Figure 3 in this paper, we don't see a big change in heavy-tailedness across iterations. This doesn't seem to agree with the theory that more heavy-tailed weights are more well-trained. Rather the main difference is between different types of weight matrices (e.g. Q,K vs FFN).

**Questions:**

- How do the estimates of pl_alpha_hill evolve over training for different layers? Do they decrease as expected by theory? Showing these plots can strengthen the argument of the paper.

---

> ### Author Response · Authors · 2025-11-22
>
> >**Q1.The simple range scaling based on estimated \alpha used in Equation (3) for learning rate seems like a heuristic that lack justification. The use of min and max alpha in the scaling might also be prone to outliers.**
>
> **A1:** We appreciate the reviewer’s concern that the range–scaling rule in Equation (3) may look heuristic and potentially sensitive to outliers in the estimated α values. **We chose this design because it is simple yet highly effective in practice**, which makes it easier for practitioners to use and adopt.
>
> - **Regarding “Equation (3) … seems like a heuristic that lacks justification”**
>
> While Equation (3) is intentionally simple, it is in fact quite stable and robust in practice. To better characterize its effect, we conducted a small sensitivity study over a range of [1,s] in Equation (3) values (lower is better):
> |Model Size|Uniform|s=1.5|s=2|s=3|s=3.5|s=4|s=4.5|s=5|
> |-|-|-|-|-|-|-|-|-|
> |60M|29.97|28.97|28.41|27.96|27.93|**27.90**|27.93|28.13|
> |135M|23.36|22.60|22.29|**22.05**|22.06|22.16|22.13|22.23|
>
> We find that setting **s in the range 3.0–4.5 yields very consistent and smooth results**, showing that this simple rule is robust without delicate tuning. For stability and ease of use, we recommend using relatively conservative values such as  s = 3.0, which perform robustly from LLaMa-60M to LLaMa-7B in our experiments.
>
> - **Regarding “the use of min and max alpha in the scaling might also be prone to outliers”**
>
> In **Figure 13 in Section D (Page 18)**, we plot the PL_Alpha_Hill of all major parameter groups over the entire training trajectory, and we observe PL_Alpha_Hill values lie in a relatively narrow range, evolve smoothly over time, and do not exhibit extreme, isolated spikes. Thus, the minimum and maximum α values used in Equation (3) are drawn from a well behaved distribution rather than being dominated by pathological outliers.
>
> >**Q2.The work by Martin & Mahoney 2019 on heavy-tailedness is mostly based on empirical results from CNN, not on LLMs. And from Figure 3 in this paper, we don't see a big change in heavy-tailedness across iterations. This doesn't seem to agree with the theory that more heavy-tailed weights are more well-trained. Rather the main difference is between different types of weight matrices (e.g. Q,K vs FFN).**
>
> **A2:** We thank the reviewer for raising this important question about the applicability of heavy‑tailed theory to LLMs and about the evolution of heavy‑tailedness in Figure 3.
>
> - **More heavy-tailed weights are more well-trained**
>
> In the revised version, we add **Figure 13 in Section D (Page 18)**, where we report PL_Alpha_Hill for all major parameter groups over the full training trajectory of LLaMA‑130M. We observe two patterns:
>
> (1) PL_Alpha_Hill evolves in a way that closely mirrors the training loss (the corresponding loss curves are shown in **Figure 14 in Section D, Page 18)**: both exhibit substantial changes in the early stage of training (roughly the first 20% of tokens), and then become much flatter later on. This synchronized evolution provides empirical evidence for the heuristic that “more heavy‑tailed weights are more well‑trained.”
>
> (2) There are systematic differences in PL_Alpha_Hill across different types of weight matrices (e.g., attention Q/K vs. FFN), which is precisely one of the key empirical findings of our paper. This heterogeneity indicates that a one‑size‑fits‑all Uniform learning‑rate scheme is unlikely to optimize all parameter groups equally well, and motivates our α‑aware LLR design.
>
> - **HT-SR findings in LLM**
>
> Recent work on LLM pruning, such as AlphaPruning [1], has already validated the usefulness of HT‑SR–based, α‑driven metrics in LLMs for pruning tasks, supporting the relevance of heavy‑tailed theory beyond CNNs. Our contribution is to show that HT‑SR is also operative in LLM pre‑training: our work includes new empirical evidence where our approach consistently surpasses all existing HT-SR methods, achieving the strongest results reported to date in the LLM pre-training setting.
>
> [1] Haiquan Lu, et al. Alphapruning: Using heavy-tailed self regularization theory for improved layer-wise pruning of large language models.

---

> ### Author Response · Authors · 2025-11-22
>
> >**Q3.How do the estimates of pl_alpha_hill evolve over training for different layers? Do they decrease as expected by theory? Showing these plots can strengthen the argument of the paper.**
>
> **A3:** We thank the reviewer for this suggestion. In the revised submission, we provide the requested plots in **Figure 13 of Section D.2 (Page 18)**. These results confirm that the pl_alpha_hill of all parameter groups decreases over training as predicted by the theory, and they also reveal several additional findings:
>
> 1) The attention parameters (Att.q, Att.k, Att.v, Att.o) consistently exhibit smaller PL_Alpha_Hill, whereas the FFN parameters (FFN.gate, FFN.up, FFN.down) maintain larger PL_Alpha_Hill throughout training.
>
> 2) The PL_Alpha_Hill of all parameter groups changes substantially during the early phase of training (approximately the first 20% of update steps). This pronounced variation highlights the importance of periodically updating the layer-wise LRs, rather than keeping them fixed over time.
>
> 3) Compared with the Uniform baseline, LLR markedly reduces PL_Alpha_Hill across all parameter groups. This reduction correlates with the improved perplexity achieved by LLR, suggesting that better control of PL_Alpha_Hill contributes directly to the enhanced training performance.

---

### Official Review · Reviewer_WV8G · 2025-10-31

**Soundness:** 4
**Presentation:** 3
**Contribution:** 4
**Rating:** 6
**Confidence:** 3

**Summary:**

This paper proposes Layerwise Learning Rate (LLR), an adaptive layer-wise learning rate strategy for large language models (LLMs), grounded in Heavy-Tailed Self-Regularization (HT-SR) theory. By periodically estimating the heavy-tailedness (PL_Alpha_Hill) of each Transformer layer’s weight or gradient spectrum, LLR assigns larger learning rates to less heavy-tailed (undertrained) layers and smaller ones to more heavy-tailed (well-trained) layers. The method transitions from gradient-based spectral estimation at early training to weight-based estimation later, reflecting the empirical evolution of spectral properties. Across LLaMA models (60M-1B) and GPT-nano with AdamW and Muon optimizers, LLR achieves up to 1.5× faster convergence and consistently lower perplexity, while requiring minimal tuning.

**Strengths:**

- First to apply HT-SR heavy-tailed spectral analysis for dynamic layerwise LR adjustment during LLM pretraining.
- Builds upon established HT-SR theory linking heavy-tailedness to training quality; the design aligns with spectral evolution trends.
- Strong, multi-scale results (60M-1B) across two optimizers, with consistent perplexity and convergence gains.
- Comprehensive studies on PL-fitting methods, update intervals, scaling ratios, and weight-gradient combinations.
- Inherits near-optimal uniform LR settings, minimizing tuning overhead.

**Weaknesses:**

- The study omits comparison to Layerwise LR Decay (LLRD) or similar depth-based heuristic schedules widely used in LLM fine-tuning.
- The paper claims low overhead, but it does not quantitatively report the computational cost of periodic heavy-tailedness estimation and PL-fitting. Since this step involves eigenvalue or singular value analysis per layer, its runtime impact should be clarified to confirm practical efficiency.
- The grad to weight transition is motivated by spectral observations and partially ablated, but a direct 3-way comparison (grad-only vs. weight-only vs. two-phase) is missing.
- The scalability of LLR beyond 1B parameters remains unclear; no evidence or discussion is provided on whether spectral estimation and PL-fitting remain stable and efficient for very large models.

**Questions:**

- Could the authors quantify the computational overhead introduced by periodic heavy-tailedness estimation and PL-fitting (e.g., as a percentage of total training time or step cost)? This is important to assess whether LLR is practically lightweight as claimed.
- Can the authors provide a direct ablation comparison among grad-only, weight-only, and two-phase versions of LLR, reporting both convergence speed and final perplexity?
- Would LLR remain effective when combined with Layerwise LR Decay (LLRD) or other depth-based heuristic schedules commonly used in practice?
- Could the authors demonstrate or discuss the scalability of LLR to larger models (e.g., 7B-70B)? If large-scale experiments are infeasible, can they provide analytical or empirical evidence that spectral estimation and PL-fitting remain computationally tractable and statistically stable as model size increases?

---

> ### Author Response · Authors · 2025-11-22
>
> >**Q1. The study omits comparison to Layerwise LR Decay (LLRD) or similar depth-based heuristic schedules widely used in LLM fine-tuning.**
>
> **A1:** Thank you for pointing out the importance of comparing against depth-based learning-rate schedules such as Layerwise LR Decay (LLRD). We have conducted additional experiments to include these baselines. Concretely, we evaluate four LLRD variants on LLaMA-60M and LLaMA-135M:
>
> - LLRD-Linear-pos: the layer-wise LR increases linearly from lr = 0.001 at the first layer to 3 × lr at the final layer;
> - LLRD-Exp-pos: the LR increases exponentially from lr = 0.001 to 3 × lr;
> - LLRD-Linear-neg: the LR decreases linearly from 3 × lr at the first layer to lr at the final layer;
> - LLRD-Exp-neg: the LR decreases exponentially from 3 × lr at the first layer to lr at the final layer;
>
> The results are as follows (lower is better):
> |Model Size|Uniform|LLRD-Linear-pos|LLRD-exp-pos|LLRD-Linear-neg|LLRD-exp-neg|LLR |
> |-|-|-|-|-|-|-|
> |60M|28.01|29.78|29.77|29.73|29.74|**27.46** |
> |135M|23.36|23.46|23.47|22.98|23.01|**22.05** |
>
> We observe that depth-based LLRD schedules do not improve over the simple uniform schedule, while our proposed LLR consistently achieves the best performance on both model sizes. For more experiments on LLRD, please refer to the answer to **A5** below.
>
> >**Q2.The paper claims low overhead, but it does not quantitatively report the computational cost of periodic heavy-tailedness estimation and PL-fitting. Since this step involves eigenvalue or singular value analysis per layer, its runtime impact should be clarified to confirm practical efficiency. Could the authors quantify the computational overhead introduced by periodic heavy-tailedness estimation and PL-fitting (e.g., as a percentage of total training time or step cost)? This is important to assess whether LLR is practically lightweight as claimed.**
>
> **A2:** Thank you for pointing this out. Clarifying the runtime is crucial for assessing the practicality of LLR. Since heavy‑tailedness estimation is only performed occasionally (every 500 steps), LLR introduces **only ≈2–3.5% runtime overhead**; meanwhile, as shown in Figure 9, it achieves up to **1.5× faster convergence** in terms of training steps to reach the same performance, easily offsetting this small overhead and keeping LLR lightweight.
>
> - **End to end training overhead.**
>
> We report the end to end training cost in H100 GPU hours for different model sizes and token budgets, with and without LLR:
> |Model size-tokens|60M-1B|130M-2B|350M-6B|1B-10B|1B-20B|7B-10B|
> |-|-|-|-|-|-|-|
> |**Uniform**|1.002|3.3| 26.28|244|516|696|
> |**LLR**|1.034 |3.4|27.12|252|528|720|
> |Added cost （%）|3.19%|3.03%|3.20%|3.28%|2.33%|3.45%|
>
> We will revise the camera ready version to make these overhead numbers more explicit so that the practical efficiency of LLR is clearer.
>
> >**Q3. The grad to weight transition is motivated by spectral observations and partially ablated, but a direct 3-way comparison (grad-only vs. weight-only vs. two-phase) is missing. Can the authors provide a direct ablation comparison among grad-only, weight-only, and two-phase versions of LLR, reporting both convergence speed and final perplexity?**
>
> **A3:** Thank you for the suggestion. We agree that a direct comparison is important. We ran an ablation on LLaMa-130M with three LRs (0.002, 0.001, 0.0008), comparing:
>
> - LLR-weight: LLR based only on weight spectra;
> - LLR-grad: LLR based only on gradient spectra;
> - LLR: our two-phase method (weight phase → grad phase).
>
> The final validation perplexities are (lower is better):
> |LR|Uniform|LLR-weight|LLR-grad|LLR (two-phase)|
> |-|-|-|-|-|
> | 0.002 |22.29| 22.11|22.14|**21.91**|
> | 0.001 |23.36| 22.27|22.30|**22.05**|
>
> Across all settings, the two-phase LLR achieves the best perplexity, while both LLR-weight and LLR-grad already improve over the uniform baseline. This confirms that (i) using spectral information from either weights or gradients is beneficial, and (ii) combining them in the proposed two-phase schedule is consistently superior. We will add this table and a brief discussion to the revised version.

---

> ### Author Response · Authors · 2025-11-22
>
> >**Q4. The scalability of LLR beyond 1B parameters remains unclear; no evidence or discussion is provided on whether spectral estimation and PL-fitting remain stable and efficient for very large models. Could the authors demonstrate or discuss the scalability of LLR to larger models (e.g., 7B-70B)? If large-scale experiments are infeasible, can they provide analytical or empirical evidence that spectral estimation and PL-fitting remain computationally tractable and statistically stable as model size increases?**
>
> **A4:** To address the concern about the evaluation scope, We have scaled up our training to more advanced settings, including 7B model, larger number of training tokens. Our new results again demonstrate the superiority of LLR.
>
> - **Pretraining LLaMa-7B with 10B tokens.**
>
> We pre-trained LLaMA-7B on 10B tokens (due to computational and rebuttal-time constraints) with LR=0.0002. The validation perplexity and zero-shot performance on common-sense reasoning benchmarks are:
> |Method|c4-ppl(↓)|ARC-c(↑)|ARC-e(↑)|Hellaswag(↑)|OBQA(↑)|PIQA(↑)|SIQA(↑)|Winogrande(↑)|Zero-shot Avg.(↑)|
> |-|-|-|-|-|-|-|-|-|-|
> |**Uniform**|17.53|18.26|46.88|30.20|15.40|65.67|36.44|51.07|37.70|
> |**LLR**|**14.20**|**21.33**|**50.63**|**34.76**|**21.20**|**68.23**|**38.64**|**51.99**| **40.97**|
>
> LLR consistently improves both perplexity and downstream zero-shot accuracy compared to the uniform baseline.
>
> - **Pretraining LLaMa-1B with 20B tokens**
>
> We further pre-trained a LLaMA-1B model on C4 by extending the training budget from 10B to 20B tokens. The results are:
> |Method|c4-ppl(↓)|ARC-c(↑)|ARC-e(↑)|Hellaswag(↑)|OBQA(↑)|PIQA(↑)|SIQA(↑)|Winogrande(↑)|Zero-shot Avg.(↑) |
> |-|-|-|-|-|-|-|-|-|-|
> |**Uniform**|14.69|20.73|48.95|34.42|18.80|68.01|39.61|**51.62**|40.30|
> |**LLR**|**13.00**|**22.61**|**51.89**|**38.36**|**21.20**|**71.06**|**39.92**|51.46|**42.36**|
>
> Across these larger-scale settings, LLR consistently outperforms the uniform baseline.
>
> >**Q5. Would LLR remain effective when combined with Layerwise LR Decay (LLRD) or other depth-based heuristic schedules commonly used in practice?**
>
> **A5:** Thank you for this question. We tested whether LLR remains effective when combined with LLRD, using both linear and exponential depth-based schedules with positive and negative decay. On 60M and 135M models, we compared a uniform LR baseline, our LLR method, and LLR combined with various LLRD schedules (see the same introductions in **A1**). The final validation perplexities are (lower is better):
> |Model Size|Uniform|LRD-Linear-pos|LLRD-exp-pos|LLRD-Linear-neg|LLRD-exp-neg|LLR|
> |-|-|-|-|-|-|-|
> |60M|28.01|29.78|29.77|29.73|29.74|**27.46**|
> |135M|23.36|23.46|23.47|22.98|23.0|**22.05**|
>
> We observe that LLR alone consistently improves over the Uniform baseline, and while adding LLRD on top of LLR does not further improve LLR itself, it does significantly improve over using LLRD alone (see results for LLRD-only in **A1**).
>
> This suggests that LLR is already well-calibrated across depth, and while it can be combined with depth-based heuristics, the main gains come from LLR itself. We will clarify this interaction and include the above table in the revised version.

---

### Official Review · Reviewer_zE9Z · 2025-11-07

**Soundness:** 2
**Presentation:** 2
**Contribution:** 2
**Rating:** 4
**Confidence:** 3

**Summary:**

This paper introduces Layerwise Learning Rate (LLR), an adaptive scheme that assigns per-layer learning rates using spectral statistics derived from heavy-tailed self-regularization theory. Building on observations that well-trained model weights exhibit heavy-tailed spectral distributions, the method dynamically adjusts each layer’s learning rate according to its degree of heavy-tailedness, promoting balanced optimization across the network. Unlike prior non-adaptive layerwise LR methods, which fail to outperform a well-tuned uniform LR, LLR periodically measures the empirical spectral density to guide adaptive updates. Experiments on Transformer models up to one billion parameters show faster step-wise convergence and modest generalization gains over standard baselines.

**Strengths:**

* **Theoretical motivation.** Builds on well-documented heavy-tailed behavior in the spectra of trained model weights and connects it to optimization dynamics.
* **Novel application to LLM pre-training.** Leverages heavy-tailed self-regularization for adaptive layerwise LR assignment—an area unexplored in large-scale Transformer training.
* **Addresses an important regime.** Seeks to improve optimization even when the global uniform LR is already near-optimal, a practically challenging and relevant setting.

**Weaknesses:**

* **Baseline coverage and tuning transparency.** Comparisons exclude recent blockwise-LR or adaptive-sharpness baselines, and tuning procedures for existing methods are under-specified.
* **Limited novelty over prior work.** The approach extends existing heavy-tailed self-regularization ideas (e.g., TempBalance, AlphaDecay) to LLMs rather than introducing a fundamentally new principle.
* **Lack of causal validation.** The link between heavy-tailedness and the “need for larger LR” is assumed but not empirically isolated through control or ablation studies (e.g., randomizing or inverting the mapping).
* **Evaluation scope is narrow.** Experiments are limited to models ≤1 B parameters on a single dataset (C4), which weakens claims about LLM-scale generality.
* **Design inconsistency.** The embedding layer is manually fixed to the maximum LR, contradicting the proposed fully data-driven adaptation rule.

**Questions:**

I suspect that self-attention layers converge faster because their hidden dimensionality is smaller than that of FFN and embedding layers, leading to a lower effective rank. Since lower effective rank typically correlates with a more skewed (possibly heavy-tailed) singular value distribution, would increasing the head dimension slow down self-attention convergence and better align its spectral characteristics with those of FFN and embedding layers?

---

> ### Author Response · Authors · 2025-11-22
>
> >**Q1.Baseline coverage and tuning transparency. Comparisons exclude recent blockwise-LR or adaptive-sharpness baselines, and tuning procedures for existing methods are under-specified.**
>
> **A1:** Thank you for raising this important point about baseline coverage and tuning transparency. In fact, we want to highlight that **we have already included the blockwise-LR baselines and tuning details in our submission.**
>
> - **Baseline coverage.**
>
> We have conducted an extensive literature review on layerwise and blockwise learning-rate schedules, including blockwise-LR [1-2] and adaptive-sharpness [3] methods. In the submitted manuscript, we explicitly include and discuss these baselines in Section 4.1 (lines 298–305). **Our experiments compare against all relevant methods we could identify from the recent literature.** If there are specific additional baselines you believe are particularly important and that we may have missed, we would be happy to include them in our experiments in the final version. Below are the results from Table 2 in our original manuscript (lower is better):
> |Model Size|Uniform|LARS[1]|LAMB[2]|Sharpness[3]|LLR|
> |-|-|-|-|-|-|
> | 60M| 28.01|36.15|30.14|29.94|**27.46**|
> | 135M| 23.36|25.33|23.30|24.29|**22.05**|
> | 350M| 16.98|17.57|17.39|19.20|**16.39**|
> | 1B| 15.60|14.22|14.13|16.46|**13.63**|
>
> - **Tuning procedure and transparency.**
>
> We have **extensively tuned the hyperparameters for all methods, with particular emphasis on carefully selecting the LR for each method**. The hyperparameter tuning process is described in detail in Figure 1 of the submitted manuscript, and the final chosen hyperparameters for all methods are reported in Tables 1, 4, 9, and 11. These tables list the key parameters (e.g., LR, weight decay, and other method-specific settings) used in our comparisons, ensuring that the tuning of both our method and the baselines is transparent and reproducible.
>
> [1] Yang You, et al. Large batch training of convolutional networks.
>
> [2] YangYou, et al. Large batch optimization for deep learning: Training bert in 76 minutes.
>
> [3] Jinbo Wang, et al. The sharpness dis parity principle in transformers for accelerating language model pre-training.
>
> >**Q2.Limited novelty over prior work. The approach extends existing heavy-tailed self-regularization ideas (e.g., TempBalance, AlphaDecay) to LLMs rather than introducing a fundamentally new principle.**
>
> **A2:** We respectfully disagree. While our method is inspired by HT-SR, our contribution is novel in both goal and technique.
>
> (1) **Novel research goal.** Prior HT-SR works focus primarily on improving optimization stability or model analysis in relatively small-scale architectures (e.g., CNNs or fine-tuning scenarios). No previous work like us investigates HT-SR as a layerwise LR solution to **solve the curse of depth [2] specifically in the context of deep, multi-billion-parameter LLM pre-training.**
>
> (2) **Novel technical contribution.**  Our method is technically distinct from TempBalance and AlphaDecay in three ways. First, whereas TempBalance is mainly developed for small vision models such as ResNet and VGG, LLR is designed for the more heterogeneous and challenging setting of LLM pre-training, and it optimizes a more critical hyper-parameter, layerwise learning rates, rather than the weight-decay schedules targeted by AlphaDecay. Second, instead of deriving HT-SR statistics solely from weights, we leverage recent advances [1] to compute HT-SR from early-stage gradients, which more faithfully capture the optimization dynamics of LLM training. Third, we introduce an LLM-specific treatment of embeddings, which are among the most influential parameters in LLMs [3] but are not explicitly handled in TempBalance or AlphaDecay; LLR tailors their learning rate and regularization, yielding substantial empirical gains.
> |LLaMa-130M-LR|Uniform|AlphaDecay|Tempbalance|LLR-weight|LLR-grad|LLR-untuned-embed|LLR|
> |-|-|-|-|-|-|-|-|
> |0.002|22.29|22.02|22.16|22.11|22.14|22.09|**21.91**|
> |0.001|23.36|22.78|23.14|22.27|22.30|22.84|**22.05**|
>
> (3) **New state of the art.** We further highlight that our work includes new empirical evidence where our approach consistently surpasses all existing HT-SR methods, achieving the strongest results reported to date in the LLM pre-training setting.
>
> In summary, while our work builds on the general HT-SR framework, it introduces both: **a new problem focus (curse of depth in LLM)**, and **a new practical technique specifically tailored and validated for large-scale LLM pre-training**. We believe these represent substantial and meaningful novelty beyond prior work.
>
> [1] Jimmy Ba, et al. High dimensional asymptotics of feature learning: How one gradient step improves the representation.
>
> [2] Wenfang Sun, et al. The curse of depth in large language models.
>
> [3] Soufiane Hayou and Liyuan Liu. Optimal embedding learning rate in llms: The effect of vocabulary size.

---

> ### Author Response · Authors · 2025-11-22
>
> >**Q3. Lack of causal validation. The link between heavy-tailedness and the “need for larger LR” is assumed but not empirically isolated through control or ablation studies (e.g., randomizing or inverting the mapping).**
>
> **A3:** Thank you for pointing this out. We have added new experiments on LLaMa-130M to directly address your concern. “Random” and “LLR-rev” denote using random layer-wise LRs and inverting the LLR mapping, respectively. In addition, we use Gradnorm, Fnorm, and Spectral as alternative indicators for controlling layer-wise LRs. Results are as follows (lower is better):
> |LR|Uniform|Random|LLR-rev|Grad-norm|F-norm|Spectral|LLR(ours)|
> |-|-|-|-|-|-|-|-|
> |0.002|22.29|22.89|54.35|22.41|32.42|32.71|**21.91**|
> |0.001|23.36|23.12|27.26|22.16|33.51|22.73|**22.05**|
> |0.0008|24.10|23.31|25.66|22.31|25.72|22.61|**22.25**|
>
> We observe that **LLR-rev consistently performs much worse than Uniform**, while our LLR consistently outperforms Uniform across all LR settings. This directly supports our claim: layers with stronger heavy-tailed spectral signatures benefit from smaller LRs, and assigning LRs in the opposite direction (LLR-rev) significantly harms model performance.
>
> >**Q4. Evaluation scope is narrow. Experiments are limited to models ≤1 B parameters on a single dataset (C4), which weakens claims about LLM-scale generality.**
>
> **A4:** To address the concern about the evaluation scope, We have scaled up our training to more advanced settings, including 7B model, larger number of training tokens, and high-quality Dolma dataset. Our new results again demonstrate the superiority of LLR.
>
> - **Pretraining LLaMa-7B with 10B tokens**
>
> We pre-trained LLaMA-7B on C4 with 10B tokens (due to computational and rebuttal-time constraints). The validation perplexity and zero-shot performance on common-sense reasoning benchmarks are:
> |Method|c4-ppl(↓)|ARC-c(↑)|ARC-e(↑)|Hellaswag(↑)|OBQA(↑)|PIQA(↑)|SIQA(↑)|Winogrande(↑)|Zero-shot Avg.(↑)|
> |-|-|-|-|-|-|-|-|-|-|
> |**Uniform**|17.53|18.26|46.88|30.20|15.40|65.67|36.44|51.07|37.70|
> |**LLR**|**14.20**|**21.33**|**50.63**|**34.76**|**21.20**|**68.23**|**38.64**|**51.99**| **40.97**|
>
> - **Pretraining LLaMa-1B with 20B tokens**
>
> We further pre-trained a LLaMA-1B model on C4 by extending the training budget from 10B to 20B tokens. The results are:
>
> |Method|c4-ppl(↓)|ARC-c(↑)|ARC-e(↑)|Hellaswag(↑)|OBQA(↑)|PIQA(↑)|SIQA(↑)|Winogrande(↑)|Zero-shot Avg.(↑) |
> |-|-|-|-|-|-|-|-|-|-|
> |**Uniform**|14.69|20.73|48.95|34.42|18.80|68.01|39.61|**51.62**|40.30|
> |**LLR**|**13.00**|**22.61**|**51.89**|**38.36**|**21.20**|**71.06**|**39.92**|51.46|**42.36**|
>
> - **Pretraining with Dolma datasets from olmo**
>
> We pre-trained olmo-tiny (135M) for 20k steps with sequence length 1024 and batch size 512 on several official Dolma datasets (MEGAWIKA, WIKIPEDIA, WIKIBOOKS, etc.). The validation perplexities on two validation sets are (lower is better):
> |Validation Dataset|Uniform|LARS|LAMB|Sharpness|LLR|
> |-|-|-|-|-|-|
> |dolma-books|19.68|22.76|19.37|19.38|**18.88**|
> |dolma-wiki|15.22|17.04|14.85|15.01|**14.40**|
>
> >**Q5.Design inconsistency. The embedding layer is manually fixed to the maximum LR, contradicting the proposed fully data-driven adaptation rule.**
>
> **A5:** Thank you for the constructive feedback. We would like to clarify that our method uses only a weight/gradient-based metric for each layer to determine the layerwise LR, and is therefore **fully data-independent**.
>
> Moreover, fixing the LR for the embedding layer is consistent with prior findings. For example, [1] shows that the embedding layer often requires separate LR tuning. However, existing methods typically rely on additional hyperparameter search to determine this LR. For instance, Adaptive-Sharpness [2] performs relatively extensive grid search to identify the optimal embedding LR.
>
> In contrast, our approach simply assigns the largest LR to the embedding layer—yielding consistent and strong performance without introducing an additional hyperparameter dimension. We believe this simplicity is **a strength: it removes the need for costly LR search while still matching or surpassing the performance of more complex methods**. Our design choice should not be penalized for being simple; rather, it offers a practical and reproducible benefit to the approach.
>
> [1] Soufiane Hayou and Liyuan Liu. Optimal embedding learning rate in llms: The effect of vocabulary size.
>
> [2] Jinbo Wang,Mingze Wang, ZhanpengZhou, JunchiYan, LeiWu, et al. The sharpness dis parity principle in transformers for accelerating language model pre-training.

---

> ### Author Response · Authors · 2025-11-22
>
> >**Q6. I suspect that self-attention layers converge faster because their hidden dimensionality is smaller than that of FFN and embedding layers, leading to a lower effective rank. Since lower effective rank typically correlates with a more skewed (possibly heavy-tailed) singular value distribution, would increasing the head dimension slow down self-attention convergence and better align its spectral characteristics with those of FFN and embedding layers?**
>
> **A6:** We thank the reviewer for this insightful question. We address this concern by examining how PL_Alpha_Hill varies with the attention head dimension, and by presenting empirical results of LLR under different head-dim settings.
>
> - **Robustness of PL_Alpha_Hill to Head Dimension**
>
> As shown in **Figure 16 in Section D.4 (page 19)** of the revised manuscript, we plot PL_Alpha_Hill for different attention head dimensions. The curves remain nearly flat across head sizes, indicating that changing the head dimension does not substantially affect the heavy‑tailedness of the corresponding weight matrices and highlighting the stability of the LLR method with respect to this architectural change.
>
> - **Robustness of LLR to Head Dimension**
>
> We applied LLR to LLaMA‑60M and LLaMA‑130M models with different attention head dimensions and compared it to the Uniform baseline. The results are (lower is better):
>
> - LLaMA 60M
>
> |head dim|32|48|64|96|192|
> |-|-|-|-|-|-|
> |**Uniform**|30.27|30.00|29.97|30.01|30.29|
> |**LLR**|**28.21**|**27.99**|**27.96**|**28.17**|**28.63**|
>
> - LLaMA 130M
>
> |head dim|48|64|96|192|768|
> |-|-|-|-|-|-|
> | **Uniform**| 23.42 | 23.36 | 23.33 | 23.46 | 24.27 |
> | **LLR** | **22.10** | **22.05** | **22.09** | **22.32** | **23.32** |
>
> Across all head dimensions, LLR consistently achieves better training performance than the Uniform baseline on both LLaMa-60M and LLaMa-130M. LLR remains robust and beneficial under different head dimensionalities.

---

### Comment · Area_Chair_FjLr · 2025-11-28

Dear Reviewers,

Thank you for your time and efforts in serving as a reviewer. The authors have submitted their rebuttal, and this AC kindly asks you to review their response and assess whether your comments have been adequately addressed.

If you have not yet done so, please raise any remaining questions by adding comments and initiating discussion as needed for points that require further clarification.

ICLR encourages reviewers to actively engage in the discussion phase, so your prompt actions are especially valuable. Thank you very much for your continued efforts and valuable contributions.

Best regards,
Your AC

---

### Author Response · Authors · 2025-12-02
**Author Response Summary**

Dear Area Chair and Reviewers,

We thank the AC and reviewers for their thoughtful efforts and overall consideration of our submission. Given our extensive and in-depth exchange with reviewers, we provide a structured summary organized into **Clarifications** on experimental configurations and novelty, **Improvements** regarding stability and baseline comparisons, and **Large-scale Experiments** (scaling to 7B), hoping this facilitates the AC’s assessment.

### **Clarifications**

**1.Hyperparameter selection & Robust gains.**

We clarify our choices of key hyperparameters, including the base Learning rate (LR) and $s$ for our method (LLR):
- **Existing LR Ranges**: Existing works on LLaMA-style pretraining with AdamW typically choose base LRs around `{60M: 0.001, 130/200M: 0.0008–0.001, 350/400M: 0.0004–0.001, 1B: 0.0002–0.0006, 7B: 0.0002}`.
- **Our LR configuration**: We use base LRs `{60M: 0.002, 135M: 0.001, 350M: 0.001, 1B: 0.0005, 7B: 0.0002}` which are deliberately kept within common ranges.
- **Stable s-range**: Setting $s$ in the range 3.0–4.5 yields consistent results, without delicate tuning.
- **Robust gains**: LLR consistently improves over the Uniform (AdamW) baseline at its optimal setting, and maintains this advantage across a broad range of LRs.
- **Overhead vs Gains**: With HT-SR estimation every 500 steps, LLR adds only 3% end‑to‑end runtime overhead but achieves up to 1.5× faster convergence.

**2.Innovations Compared to Prior Work.**

We highlight the new HT-SR–based techniques and applications in LLR:
- **LLM-tailored setting and objective**: LLR customizes layerwise LRs for large-scale LLM pre-training, unlike TempBalance(small vision models such as ResNet/VGG) and AlphaDecay(weight-decay schedules).
- **Gradient-based HT-SR statistics**:LLR is the first to derive HT-SR statistics from early-stage gradients, more faithfully capturing the optimization dynamics of LLM training.
- **LLM-specific embedding treatment**: LLR leverages HT-SR findings to tailor the LRs of embeddings—key LLM parameters ignored by TempBalance and AlphaDecay—yielding substantial empirical gains.

### **Improvements**
**3.Stability and Robustness of LLR.**

LLR is stable and consistently beneficial across architectures, indicators, and training regimes.
  - **Stable across sizes**: From LLaMa-60M to LLaMa-7B, LLR consistently outperforms the Uniform (AdamW) baseline.
  - **Broader frameworks and datasets**: On Olmo‑tiny (135M) with Dolma (books/wiki), LLR consistently beats AdamW/Uniform and other baselines.
  - **Validated mapping**: On LLaMA‑130M, random LRs, reversed LLR (LLR‑rev), and alternative indicators (Grad‑norm, F‑norm, Spectral) all underperform LLR.
  - **Defeating loss spike**: Using a [0.33, 1] scaling range, LLR remains stable and strong even at very large global LRs, effectively suppressing loss spikes where Uniform diverges.
  - **Robust to head dimension**: PL_Alpha_Hill is nearly invariant to head size, and across all head dims on LLaMA‑60M/130M, LLR consistently outperforms the Uniform.

**4.Additional comparisons to related methods.**

We provide more comparisons to existing adaptive layerwise optimization methods:
- **Layerwise LR Decay(LLRD)**: On LLaMA‑60M/135M, all four LLRD variants fail to outperform the Uniform baseline, while LLR consistently achieves the best perplexity.
- **AlphaDecay/TempBalance**: On LLaMA‑130M, AlphaDecay and TempBalance yield only modest gains over Uniform, while LLR achieves clearly better perplexity in all settings.

### **Large-scale experiments**

We have conducted large-scale experiments, confirming that LLR remains robust across model scales, token budgets, and evaluation tasks:

- **Scaling to LLaMA-7B**: For LLaMA-7B trained on 10B tokens, LLR outperforms Uniform by reducing validation PPL (14.20 vs 17.53).
- **Robust over tokens**: Across different token budgets, LLR consistently outperforms Uniform—for example on the 1B model from 10B to 20B tokens, and on the 130M model from 2B to 32B tokens.
- **Better zero-shot**: LLR improves zero-shot generalization—for example, the 1B model with 20B tokens scores 42.36 vs 40.30 for Uniform on common-sense benchmarks.

We hope our clarifications help to highlight LLR’s contribution: introducing HT-SR–driven layerwise LR techniques that enable more stable optimization, better data efficiency, and new state-of-the-art LLM pretraining performance across model scales and token budgets.

---

### Meta-Review · Area_Chair_YcTf · 2026-01-07

**Summary:**

The reviewers were split about this paper and did not come to a consensus. On one hand they appreciated the size of the evaluation and the structure of the paper. On the other they had concerns with (a) the theory of the paper, (b) the baselines used, (c) the sensitivity of hyperparameters, (d) suspicious results. The authors were unable to adequately respond to these concerns. In fact a reviewer was inclined to decrease their score because of the rebuttal. Because of this, I vote to reject.

**Reviewer Concerns:**

Please see above.

**Reviewer Scores:**

I believe reviewers would have lowered their scores or kept them the same.

---

### Decision · Program_Chairs · 2026-01-26

Reject